# Self-Attention Between Datapoints: Going Beyond Individual Input-Output Pairs in Deep Learning

Jannik Kossen[1]*        Neil Band[1]*

Clare Lyle[1]   Aidan N. Gomez[1,3]   Tom Rainforth[2]   Yarin Gal[1]

[1] OATML, Department of Computer Science, University of Oxford
[2] Department of Statistics, University of Oxford
[3] Cohere

## Abstract

We challenge a common assumption underlying most supervised *deep learning*: that a model makes a prediction depending only on its parameters and the features of a *single input*. To this end, we introduce a general-purpose deep learning architecture that takes as input the *entire dataset* instead of processing one datapoint at a time. Our approach uses self-attention to reason about relationships between datapoints explicitly, which can be seen as realizing non-parametric models using parametric attention mechanisms. However, unlike conventional non-parametric models, we let the model learn end-to-end from the data how to make use of other datapoints for prediction. Empirically, our models solve cross-datapoint lookup and complex reasoning tasks unsolvable by traditional deep learning models. We show highly competitive results on tabular data, early results on CIFAR-10, and give insight into how the model makes use of the interactions between points.

## 1 Introduction

From CNNs [57] to Transformers [90], most of supervised deep learning relies on *parametric* modeling: models learn parameters $\boldsymbol{\theta}$ from a set of training data $\mathcal{D}_{\text{train}} = \{(\boldsymbol{x}_1, \boldsymbol{y}_1), \ldots, (\boldsymbol{x}_n, \boldsymbol{y}_n)\}$ to maximize training likelihoods $p(\boldsymbol{y} \mid \boldsymbol{x}; \boldsymbol{\theta})$ mapping from features $\boldsymbol{x} \in \mathcal{X}$ to target values $\boldsymbol{y} \in \mathcal{Y}$. At test time, they then make a prediction $p(\boldsymbol{y}^* \mid \boldsymbol{x}^*; \boldsymbol{\theta})$ that depends only on those parameters $\boldsymbol{\theta}$ and the test input $\boldsymbol{x}^*$. That is, parametric models do not consider direct dependencies between datapoints.

This paper challenges parametric modeling as the dominant paradigm in deep learning. Based on the same end-to-end learning motivations that underpin deep learning itself, we consider giving models the *additional flexibility* of using training data *directly* when making predictions $p(\boldsymbol{y}^* \mid \boldsymbol{x}^*, \mathcal{D}_{\text{train}}; \boldsymbol{\theta})$.

Concretely, we introduce **Non-Parametric Transformers (NPTs)**: a general deep learning architecture that takes the entire dataset as input and predicts by explicitly *learning* interactions between datapoints (Fig. 1). NPTs leverage both parametric and *non*-parametric predictive mechanisms, with the use of end-to-end training allowing the model to naturally learn from the data how to balance the two. Namely, instead of just learning predictive functions from the features to the targets of independent datapoints, NPTs can also learn to reason about general relationships *between* inputs. We use multi-head self-attention [4, 59, 90] to model relationships between datapoints and construct

---

***Equal Contribution.** Correspondence to {jannik.kossen, neil.band}@cs.ox.ac.uk.

35th Conference on Neural Information Processing Systems (NeurIPS 2021).

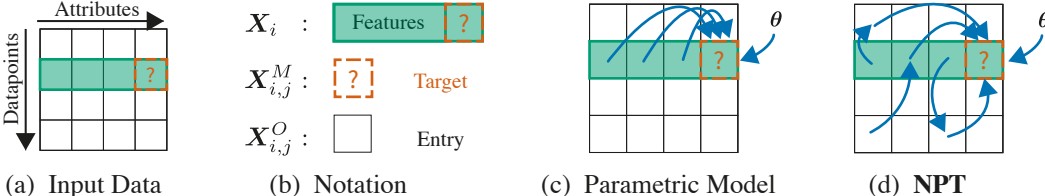

Figure 1: NPTs learn direct interactions between datapoints. (a) Input data: predict masked target entry [?] for datapoint $X_i$. (b) Notation from §2. (c) Parametric models predict only from the features of the given input. (d) NPTs predict by modeling relationships between all points in the dataset.

a training objective for NPTs with a stochastic masking mechanism inspired by self-supervised reconstruction tasks in natural language processing [24]. We show that these models *learn* to look up information from other datapoints and capture the causal mechanism generating the data in semi-synthetic settings. However, unlike conventional non-parametric models, NPTs are not forced to *only* make predictions in this way: they can also use the power of ordinary parametric deep learning.

**Background.** While questioning parametric modeling assumptions is unconventional in deep learning, in statistics, so-called *non-parametric* models are a well-known and long-established field of study. Non-parametric models make predictions in explicit dependence of the training data $p(y^* \mid x^*, \mathcal{D}_{train})$. The most popular example of such models in the machine learning community are perhaps Gaussian Processes [74]. Non-parametric models typically do not require any training of parameters, and instead often directly interpolate between training points according to a fixed procedure, e.g., [74, p.17]. The interactions between inputs are fully defined by architectural choices and a small set of hyperparameters that must be carefully chosen. Conventional non-parametric models cannot *learn* – in the sense familiar to deep learning practitioners – interactions from the data, limiting the flexibility these models have in adapting to the data at hand. Approaches such as Deep Gaussian Processes [22], Deep Kernel Learning [95], and Neural Processes [36, 37, 49] have all sought to apply ideas from deep neural networks to non-parametrics. Compared to NPTs, these approaches rely heavily on motivations from stochastic processes. This leads to them being either less flexible than NPTs or requiring strong assumptions on the data, making them *inapplicable* to the practical scenarios considered in this paper (cf. §3). Unlike previous work, NPTs explicitly learn to predict from interactions between datapoints, and they can be applied to general supervised machine learning tasks. We refer to §3 for an overview of these and other related approaches.

A key contribution of this paper is opening the door to a more general treatment of how deep learning models can make use of dependencies between datapoints for predictions. Our results demonstrate that NPTs make use of interactions between datapoints in practice, and we show highly competitive performance on several established tabular datasets as well as early image classification results. Additionally, we show that NPTs can solve complex reasoning tasks by combining representation learning and cross-datapoint lookup; something that is impossible for conventional deep learning or non-parametric models due to their inability to *learn* relations *between* datapoints.

We next discuss the specifics of our model (§2), before moving on to related work (§3), empirical results (§4), and finally, limitations, future work, and conclusions (§5).

## 2 Non-Parametric Transformers

Non-Parametric Transformers (NPTs) explicitly *learn* relationships between datapoints to improve predictions. To accomplish this, they rely on three main ingredients: **(1)** We provide the model with the **entire dataset – all datapoints – as input**. We approximate this with minibatches where necessary for large data. At test time, both training and test data are input to the model; during training, the model learns to predict targets from the training data (§2.6). **(2)** We use **self-attention between datapoints** to explicitly model relationships between datapoints. For example, at test time, the attention mechanism models relationships amongst training points, amongst test points, and between the two. **(3)** NPT's training objective is to reconstruct a corrupted version of the input dataset. Similar to BERT [24], we apply **stochastic masking** to inputs and minimize a loss on predictions at entries masked out in the input. Next, we introduce the three components in detail.

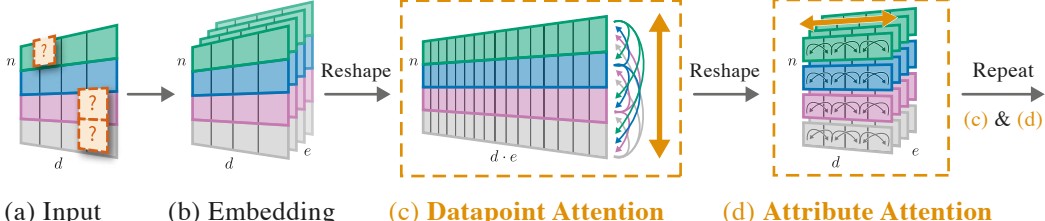

| (a) Input | (b) Embedding | (c) **Datapoint Attention** | (d) **Attribute Attention** |

Figure 2: Overview of the Non-Parametric Transformer. (a) The input dataset and mask matrix are stacked and (b) linearly embedded for all datapoints independently. NPT then applies (c) Attention Between Datapoints (ABD, §2.4) across all $n$ samples of hidden dimension $h = d \cdot e$. (d) Attention Between Attributes (ABA, §2.5) then attends between the attributes for each datapoint independently. We repeat steps (c) and (d) and obtain a final prediction from a separate linear projection (not shown).

## 2.1  Datasets as Inputs

NPTs take as input the entire dataset $\boldsymbol{X} \in \mathbb{R}^{n \times d}$. The datapoints are stacked as the rows of this matrix $\{\boldsymbol{X}_{i,:} \in \mathbb{R}^d \mid i \in 1 \ldots n\}$, and we refer to the columns as attributes $\{\boldsymbol{X}_{:,j} \in \mathbb{R}^n \mid j \in 1 \ldots d\}$. Each attribute is assumed to share a semantic meaning among all datapoints. In single-target classification and regression, we assume that the targets (labels) are the final attribute $\boldsymbol{X}_{:,d}$, and the other attributes $\{\boldsymbol{X}_{:,j} \mid j \neq d\}$ are input features, e.g., the pixels of an image. Each $\boldsymbol{X}_{i,j}$ is an entry or value. In addition to tabular data, many modalities such as images, graphs, or timeseries can be reshaped to fit this format. Note that this is a departure from common notation for supervised learning as introduced in §1, as the input $\boldsymbol{X}$ now includes both features and targets (collectively, attributes).

In masked language modeling [24], mask tokens denote which words in a sentence are unknown and where, at training time, model predictions will have a loss backpropagated. Analogously, we use a binary matrix $\boldsymbol{M} \in \mathbb{R}^{n \times d}$ to specify which entries are *masked* in the input $\boldsymbol{X}$. This matrix is also passed to NPT as input. The task is to predict the masked values $\boldsymbol{X}^M = \{\boldsymbol{X}_{i,j} \mid \boldsymbol{M}_{i,j} = 1\}$ from the observed values $\boldsymbol{X}^O = \{\boldsymbol{X}_{i,j} \mid \boldsymbol{M}_{i,j} = 0\}$, i.e., to predict $p(\boldsymbol{X}^M \mid \boldsymbol{X}^O)$.

In summary, NPT takes as input the entire dataset and masking matrix $(\boldsymbol{X}, \boldsymbol{M})$, and makes predictions $\hat{\boldsymbol{X}} \in \mathbb{R}^{n \times d}$ for values masked at input. This general setup accommodates many machine learning settings simply by adjusting the placement of the binary masks in $\boldsymbol{M}$. We focus on single-target classification and regression – corresponding to a masking matrix $\boldsymbol{M}$ with 1s at all entries of the label column $\boldsymbol{X}_{:,d}$ – but outline multi-target settings, imputation, self-supervision using input features, and semi-supervision in Appendix C.4. Next, we describe the NPT architecture.

## 2.2  NPT Architecture

An overview of the Non-Parametric Transformer (NPT) is depicted in Fig. 2. NPT receives the dataset and masking matrix $(\boldsymbol{X}, \boldsymbol{M})$ as input (Fig. 2a). We stack these and apply an identical linear embedding to each of $n$ datapoints, obtaining an input representation $\boldsymbol{H}^{(0)} \in \mathbb{R}^{n \times d \times e}$ (Fig. 2b). Next, we apply a sequence of multi-head self-attention layers [4, 24, 90]. Crucially, we alternately apply attention between *datapoints* and attention between *attributes* of individual datapoints (Figs. 2c-d).

These operations allow our model to learn both relationships between datapoints as well as transformations of individual datapoints. Finally, an output embedding gives the prediction $\hat{\boldsymbol{X}} \in \mathbb{R}^{n \times d}$, which now has predicted values at entries that were masked at input. We refer to Appendix C.3 for details, such as treatment of categorical and continuous variables. Importantly:

**Property 1.** *NPTs are equivariant to a permutation of the datapoints. (cf. Appendix A for proof.)*

In other words, if the set of input datapoints is shuffled, NPTs produce the same prediction but shuffled in an analogous manner. This explicitly encodes the assumption that the learned relations between datapoints should not depend on their ordering. At a high level, permutation-equivariance holds because all components of NPTs are permutation-equivariant, and the composition of permutation-equivariant functions is itself permutation-equivariant. We now briefly recap multi-head self-attention which plays an important role throughout the NPT architecture.

## 2.3 Multi-Head Self-Attention

Multi-head self-attention (MHSA) is a powerful mechanism for learning complex interactions between elements in an input sequence. Popularized in natural language processing [4, 24, 90], MHSA-based models have since been successfully applied to many areas of machine learning (cf. §3).

*Dot-product attention* computes attention weights by comparing queries $\{\boldsymbol{Q}_i \in \mathbb{R}^{1 \times h_k} \mid i \in 1 \ldots n\}$ with keys $\{\boldsymbol{K}_i \in \mathbb{R}^{1 \times h_k} \mid i \in 1 \ldots m\}$, ultimately updating the representation of the queries by aggregating over values $\{\boldsymbol{V}_i \in \mathbb{R}^{1 \times h_v} \mid i \in 1 \ldots m\}$ via the attention weights. We stack the queries, keys, and values into matrices $\boldsymbol{Q} \in \mathbb{R}^{n \times h_k}$, $\boldsymbol{K} \in \mathbb{R}^{m \times h_k}$, and $\boldsymbol{V} \in \mathbb{R}^{m \times h_v}$ and, as is commonly done for convenience, assume $h_k = h_v = h$. Then, we compute dot-product attention as

$$\text{Att}(\boldsymbol{Q}, \boldsymbol{K}, \boldsymbol{V}) = \text{softmax}(\boldsymbol{Q}\boldsymbol{K}^T/\sqrt{h})\boldsymbol{V}. \tag{1}$$

*Multi-head* dot-product attention concatenates a series of $k$ independent *attention heads*

$$\text{MHAtt}(\boldsymbol{Q}, \boldsymbol{K}, \boldsymbol{V}) = \underset{\text{axis}=h}{\text{concat}}(\boldsymbol{O}_1, \ldots, \boldsymbol{O}_k)\boldsymbol{W}^O, \text{ where} \tag{2}$$

$$\boldsymbol{O}_j = \text{Att}(\boldsymbol{Q}\boldsymbol{W}_j^Q, \boldsymbol{K}\boldsymbol{W}_j^K, \boldsymbol{V}\boldsymbol{W}_j^V). \tag{3}$$

We learn embedding matrices $\boldsymbol{W}_j^Q, \boldsymbol{W}_j^K, \boldsymbol{W}_j^V \in \mathbb{R}^{h \times h/k}, j \in \{1, \ldots, k\}$ for each head $j$, and $\boldsymbol{W}^O \in \mathbb{R}^{h \times h}$ mixes outputs from different heads. Here, we focus on multi-head *self*-attention, $\text{MHSelfAtt}(\boldsymbol{H}) = \text{MHAtt}(\boldsymbol{Q} = \boldsymbol{H}, \boldsymbol{K} = \boldsymbol{H}, \boldsymbol{V} = \boldsymbol{H})$, which uses the *same* inputs for queries, keys, and values. Following Transformer best practices to improve performance [16, 24, 59, 66, 90], we first add a residual branch and apply Layer Normalization (LN) [3] followed by $\text{MHSelfAtt}(\cdot)$,

$$\text{Res}(\boldsymbol{H}) = \boldsymbol{H}\boldsymbol{W}^{\text{res}} + \text{MHSelfAtt}(\text{LN}(\boldsymbol{H})), \tag{4}$$

with learnable weight matrix $\boldsymbol{W}^{\text{res}} \in \mathbb{R}^{h \times h}$. Then, we add another residual branch with LN and a row-wise feed-forward network (rFF), finally giving the full multi-head self-attention layer as

$$\text{MHSA}(\boldsymbol{H}) = \text{Res}(\boldsymbol{H}) + \text{rFF}(\text{LN}(\text{Res}(\boldsymbol{H}))) \in \mathbb{R}^{n \times h}. \tag{5}$$

## 2.4 Attention Between Datapoints (ABD)

The **Attention Between Datapoints (ABD)** layer is a key operation for NPT. It explicitly transforms data by reasoning about pairwise relationships between all datapoints, see Fig. 2c. As input to ABD, we flatten the output of the previous layer $\boldsymbol{H}^{(\ell)}$ from $\mathbb{R}^{n \times d \times e}$ to $\mathbb{R}^{n \times h}$ with $h = d \cdot e$. Then, we apply $\text{MHSA}(\cdot)$ between the intermediate datapoint representations $\{\boldsymbol{H}_i^{(\ell)} \in \mathbb{R}^{1 \times h} \mid i \in 1 \ldots n\}$ as

$$\text{ABD}(\boldsymbol{H}^{(\ell)}) = \text{MHSA}(\boldsymbol{H}^{(\ell)}) = \boldsymbol{H}^{(\ell+1)} \in \mathbb{R}^{n \times h}. \tag{6}$$

At the first ABD layer, we input $\boldsymbol{H}^{(0)} \in \mathbb{R}^{n \times d \times e}$, the linearly embedded input data. After applying ABD, we reshape the output again, from $\mathbb{R}^{n \times h}$ to $\mathbb{R}^{n \times d \times e}$. Here, the rFF of each ABD layer is an MLP that is applied independently to each of the $n$ datapoints.

Note that this is distinct from how $\text{MHSA}(\cdot)$ is usually applied in the literature, as we compute attention between *different datapoints* and not between the *features of a single datapoint* [24, 25, 46, 90]. For example, in natural language processing, attention is usually applied between the tokens (attributes) of a sentence (datapoint) but not between different sentences. For example, NPT could learn to attend between two datapoints with indices $i$ and $i'$ by embedding $\boldsymbol{Q}_i$ and $\boldsymbol{K}_{i'}$ in close proximity. Following (1), datapoint $i$ will then attend more closely to $i'$ because $\boldsymbol{Q}_i \boldsymbol{K}_{i'}^T$ will be large. By stacking many ABD layers, NPT can learn higher-order interactions between datapoints [24, 90].

## 2.5 Attention Between Attributes (ABA)

We now introduce **Attention Between Attributes (ABA)**, which we by default perform after each ABD layer. ABA layers can help the model learn better per-datapoint representations for the between-datapoint interactions, see Fig. 2d. For ABA, we apply $\text{MHSA}(\cdot)$ independently to each row (corresponding to a single datapoint) in the input $\boldsymbol{H}_i^{(\ell)} \in \mathbb{R}^{d \times e}, i \in \{1, \ldots, n\}$, giving

$$\text{ABA}(\boldsymbol{H}^{(\ell)}) = \underset{\text{axis}=n}{\text{stack}}(\text{MHSA}(\boldsymbol{H}_1^{(\ell)}), \ldots, \text{MHSA}(\boldsymbol{H}_n^{(\ell)})) = \boldsymbol{H}^{(\ell+1)} \in \mathbb{R}^{n \times d \times e}. \tag{7}$$

Just like in standard Transformers [24, 25, 46, 90], ABA is used to transform attribute representations of single datapoints independently. We batch over the $n$ dimension to compute ABA efficiently. By alternating between attention between datapoints (ABD) and attributes (ABA), NPTs can model both complex dependencies between points as well as learn suitable transformations of datapoints individually. Next, we describe the use of masking mechanisms during NPT training and evaluation.

## 2.6 Masking and Optimization

**Masking.** Much like in masked language modeling [24], we use masks to indicate which values NPT is expected to predict, and to prevent the model from accessing ground truth values. Recall that NPT needs to predict $p(\boldsymbol{X}^M \mid \boldsymbol{X}^O)$, with masked values $\boldsymbol{X}^M = \{\boldsymbol{X}_{i,j} \mid \boldsymbol{M}_{i,j} = 1\}$ and observed values $\boldsymbol{X}^O = \{\boldsymbol{X}_{i,j} \mid \boldsymbol{M}_{i,j} = 0\}$. Masked values can be either features or targets. Canonically, masked language modeling is used to perform self-supervised learning on a sequence of tokens in a sentence [24]. We use such *stochastic feature masking* to mask feature values $\boldsymbol{X}_{i,j}, j \neq d$, with probability $p_{\text{feature}}$ during training. We also apply stochastic masking to the targets of the training set $\boldsymbol{X}_{:,d}$ with probability $p_{\text{target}}$. We call this *stochastic target masking*. Note that we take great care to avoid test set leakage and *never* reveal targets of the test set to NPT. We refer to Appendix C.4 for full details of our masking procedure in a variety of settings.

**NPT Objective.** During training, we compute the negative log-likelihood loss at training targets $\mathcal{L}^{\text{Targets}}$ as well as the auxiliary loss from masked-out features $\mathcal{L}^{\text{Features}}$. We write the NPT training objective as $\mathcal{L}^{\text{NPT}} = (1 - \lambda)\mathcal{L}^{\text{Targets}} + \lambda\mathcal{L}^{\text{Features}}$, where $\lambda$ is a hyperparameter. At test time, we only mask and compute a loss over the targets of test points. See Appendix C.5 for optimization details.

This objective has a few notable elements. Feature masking requires NPTs to make predictions over all attributes, encouraging the models to learn a representation of the entire dataset. This increases the difficulty of the task and adds more supervision, which we find tends to have a beneficial regularizing effect. Interestingly, stochastic *target* masking means that many training targets are *unmasked* to the model at training time. This allows NPTs to learn to predict the masked targets of certain training datapoints using the *targets of other training datapoints* in addition to all input features.[2] NPTs no longer have to memorize a mapping between training inputs and outputs in their parameters $\boldsymbol{\theta}$, and can instead use their representational capacity to learn functions using other *training features and targets as input*. For example, NPTs could learn to assign test datapoints to clusters of training datapoints, and predict on those points using interpolation of the training targets in their respective cluster. We explore the ability of NPTs to solve such tasks in §4.2. Further, we study more complex extensions to these tasks, which cannot be solved by simple interpolative models, in Appendix B.1.2.

**Handling Large Datasets.** Due to the poor $\mathcal{O}(n^2)$ time and space complexity of self-attention, we resort to approximations once the data grows too large. For example, we reach 24 GB of GPU memory for standard NPT model sizes at about 8000 datapoints. We find that processing the data in random subsets for model training and prediction, i.e., *minibatching*, is a simple and effective solution. We construct minibatches such that, at test time, training and test data are both present in the same batch, to allow NPTs to attend to training datapoints. In §4.3, we show that NPTs make use of attention between datapoints with minibatching enabled. See §5 for further discussion and ideas for future work.

## 3 Related Work

**Deep Non-Parametric Models.** Deep Gaussian Processes [22] and Deep Kernel Learning (DKL) [95] extend ideas from Gaussian Processes [74] to representation learning. Deep GPs stack standard GPs with the aim to learn more expressive relationships between input points, sharing motivation with NPTs. However, unlike NPTs, deep GPs are difficult to work with in practice, requiring complex approximate inference schemes [13, 21, 77]. DKL applies a neural network to each datapoint *independently* before passing points on to a standard Gaussian Process, making predictions based directly on similarity in embedding space instead of *learning* the interactions themselves.

**Neural Processes.** Similar to GPs, Neural Processes (NPs) [36, 37] define a distribution over functions. They use a latent variable model parametrized by neural networks, fulfilling specific

---

[2]A concern here could be that the model will memorize training targets and fail to generalize. In practice, we do not observe generalization issues, likely because (i) a loss is never backpropagated on an unmasked value, and (ii) BERT-style masking [24] uses token randomization to prevent memorization. See Appendix C.4.

architectural constraints to approximately preserve consistency of finite-dimensional marginals. Attentive Neural Processes (ANPs) [49] extend Neural Processes to allow for direct attention between a context set and targets. However, as the authors themselves stress, "NPs and GPs have different training regimes" [49]. While a GP can be trained on a single dataset, *(A)NPs require multiple realizations of the dataset*. The authors further note that *"a direct comparison between the two is usually not plausible"* [49], which is why we cannot compare (A)NPs to NPTs on our standard tasks.

**Attention.** NPTs are part of a line of recent work that explores the use of Transformer-based architectures outside of natural language processing, e.g., Transformers in computer vision [25, 46, 67] or architectures exploiting desirable invariances or equivariances [33, 44, 59, 61]. Like NPTs, Set Transformer [59] attends to a set of input points. However, unlike NPTs, Set Transformer relies on the existence of multiple independent sets for training and makes only a single prediction for each set. Like NPTs, Axial Transformers [42] and MSA Transformers [73] attend to multiple dimensions of matrix-shaped input. However, Axial Transformers process single images as input, i.e., no attention across datapoints is performed. MSA Transformers use attention within individual protein sequences and across an aligned protein family for contact prediction, but do not consider a more general setting. Recent works have improved neural network performance on tabular data using attention. AutoInt [80] is a direct application of multi-head attention to tabular data, and TabNet [2] sequentially attends to sparse subsets of the features inspired by tree-based models. Both approaches do not reason about interactions between datapoints, a key contribution that we introduce with NPT in this work.

**Few-Shot Learning, Meta-Learning, and Prompting.** In §4.2, we apply NPTs to tasks that require learning of relational structure between datapoints on training data to achieve good generalization performance on novel test inputs. This setup shares motivations with meta-learning [6, 8, 29, 56], in which a model is pre-trained on a variety of tasks, such that it can then learn new tasks using only a small number of additional training points from the new task. However, we consider evaluation without any additional gradient updates, unlike recent meta-learning methods [29, 97] which are therefore inapplicable to this setting. Recent works on few-shot learning with text prompting [12, 72] provide a trained Transformer-based language model with a few examples of a novel relationship in a prompt at prediction time, where they observe strong generalization on the task. Similarly, we consider attention between a "context" of datapoints. While ground-truth input-output pairs are provided for prompting, we consider settings in which no ground-truth is given at prediction time (cf. Appendix B.1.2), but the model can solve the task if it has learned the underlying relational structure.

**Semi-Supervised Learning and Graph Neural Networks.** NPTs relate to work on semi-supervised learning [15, 27, 51] and transductive learning [89], which both make use of unlabeled inputs during training. NPTs natively support this by simply including any unlabeled datapoints with masked-out targets in the input matrix at training time. This body of related work includes semi-supervised and transductive learning on graphs using graph neural networks (GNNs), e.g., [34, 52, 53, 91, 96]. NPTs can be seen as a generalization of GNNs in which a set of dependencies (edges) between datapoints is not known a priori and is instead learned from data using self-attention. Like NPTs, Neural Relational Inference (NRI) [53] attempts to discover relations amongst datapoints. However, NRI lacks scalability because it requires that embeddings be stored for each potential graph edge.

**Metric Learning**. (Deep) Metric Learning aims to learn distance functions such that the (semantic) similarity and dissimilarity between input points is meaningfully captured, e.g., [65, 76, 79, 92–94]. Similarly, retrieval models in NLP learn to look up relevant training instances for prediction [38, 39, 41]. The attention between datapoints in NPTs can be seen as implicitly learning exactly such (dis-)similarity. Usually, metric learning embeds inputs by applying the same embedding function independently to each datapoint. This is in contrast to NPTs, which leverage a learned self-attention mechanism between test inputs and training datapoints (including their labels) at prediction time.

## 4 Experiments

We seek to answer the following set of questions in our evaluation[3] of NPTs: (**Q1**) How do NPTs perform on standard benchmarks for supervised machine learning? (**Q2**) Can NPTs successfully model interactions between datapoints in idealized settings? (**Q3**) Do NPTs actually learn to rely on interactions between datapoints for prediction on real-world datasets? (**Q4**) If so, what is the nature of these interactions, e.g., which other datapoints are relevant for prediction?

---

[3]We release code for NPTs at github.com/OATML/Non-Parametric-Transformers.

Table 1: Average rank order of various methods ($\pm$ standard error) on UCI benchmarks, across binary classification, multi-class classification, and regression tasks. We determine rank using the test area under the receiver operating characteristic (AUROC) curve on binary classification (4 of 10 datasets), accuracy on multi-class classification (2 of 10), and root mean squared error (RMSE) on regression (4 of 10), and sort methods by ascending rank for each metric. See Appendix B.7 for the full results.

| Method | AUROC | Method | Accuracy | Method | RMSE |
|---|---|---|---|---|---|
| NPT | **2.50 $\pm$ 0.87** | NPT | **2.50 $\pm$ 0.50** | CatBoost | **3.00 $\pm$ 0.91** |
| CatBoost | 2.75 $\pm$ 0.85 | XGBoost | **2.50 $\pm$ 1.50** | XGBoost | 3.25 $\pm$ 0.63 |
| LightGBM | 3.50 $\pm$ 1.55 | MLP | 3.00 $\pm$ 2.00 | NPT | 3.25 $\pm$ 1.31 |
| XGBoost | 4.75 $\pm$ 1.25 | CatBoost | 3.50 $\pm$ 0.50 | Gradient Boosting | 4.00 $\pm$ 1.08 |
| Gradient Boosting | 5.00 $\pm$ 0.71 | Gradient Boosting | 3.50 $\pm$ 1.50 | Random Forest | 4.50 $\pm$ 0.87 |
| MLP | 5.75 $\pm$ 1.49 | Random Forest | 6.50 $\pm$ 0.50 | MLP | 5.00 $\pm$ 1.22 |
| Random Forest | 6.00 $\pm$ 0.71 | TabNet | 7.50 $\pm$ 0.50 | LightGBM | 6.50 $\pm$ 1.55 |
| TabNet | 6.50 $\pm$ 1.32 | LightGBM | 7.50 $\pm$ 1.50 | TabNet | 6.75 $\pm$ 0.95 |
| k-NN | 8.25 $\pm$ 0.48 | k-NN | 8.50 $\pm$ 0.50 | k-NN | 8.75 $\pm$ 0.25 |

### 4.1 NPTs Perform Competitively on Established Benchmarks

To answer **(Q1)**, we evaluate NPTs on tabular data from the UCI Repository [26] as well as the CIFAR-10 [55] and MNIST [58] image classification datasets. Tabular data is ubiquitous in real-world machine learning [20] but notoriously challenging for general purpose deep neural networks, which are rarely used in practice here because they are consistently outperformed by boosting models [78].[4]

**Tabular Datasets, Setup, and Baselines.** We evaluate NPTs over 10 datasets varying across the number of datapoints, number of features, composition (categorical or continuous) of features, and task. 4 of the 10 are binary classification, 2 are multi-class classification, and 4 are regression. We compare NPT against a wide set of standard or state-of-the-art baselines: Random Forests [10], Gradient Boosting Trees [32], XGBoost [17], CatBoost [71], LightGBM [48], MLPs, k-NN [1, 30], and TabNet [2]. For additional background on tree-based models, see Appendix D.1. We tune the parameters of all models on validation sets and use 10-fold cross-validation whenever computationally feasible. Note that while we perform an extensive grid search for the baselines, we only search over a small set of configurations for NPTs. We refer the reader to Appendix E for further details on the setup for datasets and baselines, and Appendix C.1 for NPT hyperparameters.

**Tabular Data Results.** We report the average rank order for NPT and various tree-based and deep learning baselines in Table 1. NPT achieves the highest average ranking on binary and multi-class classification tasks, outperforming CatBoost and XGBoost, two popular state-of-the-art boosting methods designed specifically for tabular data. On regression tasks, NPT ties in average rank with XGBoost, and is outperformed only by CatBoost. In addition to its strong rank-wise performance, NPT achieves best performance on 4 of the 10 benchmark datasets – more than any other method. We find that these are remarkable results for a general purpose model that does not include tabular-specific design, supporting our hypothesis that attention between datapoints is a useful architectural inductive bias for prediction. For all metrics across all datasets, i.e., NLL for classification, AUROC/accuracy for binary/multi-class classification, and (R)MSE for regression, we refer the reader to Appendix B.7. In the appendix, we present ablations which suggest that the performance of NPT is robust across a wide range of hyperparameter choices (Appendix B.4) and that both the introduction of the ABA layer and the stochastic feature masking contribute positively to the performance of NPTs (Appendix B.5).

**Image Data Results.** On CIFAR-10, we replace our linear encoder with a CNN followed by ABD layers on the CNN encodings, achieving a test accuracy of 93.7%. We achieve 98.3% accuracy on MNIST using linear patching [25]. Crucially, we show in §4.3 that NPTs learn to make use of interactions between images on both the CIFAR-10 and MNIST datasets, supporting the claim that attention between datapoints is useful beyond tabular data. We also explore linear patching on CIFAR-10. See Appendix B.8 for these results along with setup details and further discussion.

---

[4]We conduct an informal survey of all Kaggle [45] competitions using tabular data completed in 2020 with a public leaderboard. In 11 out of a total of 13 cases, the winning entries relied on some form of boosting.

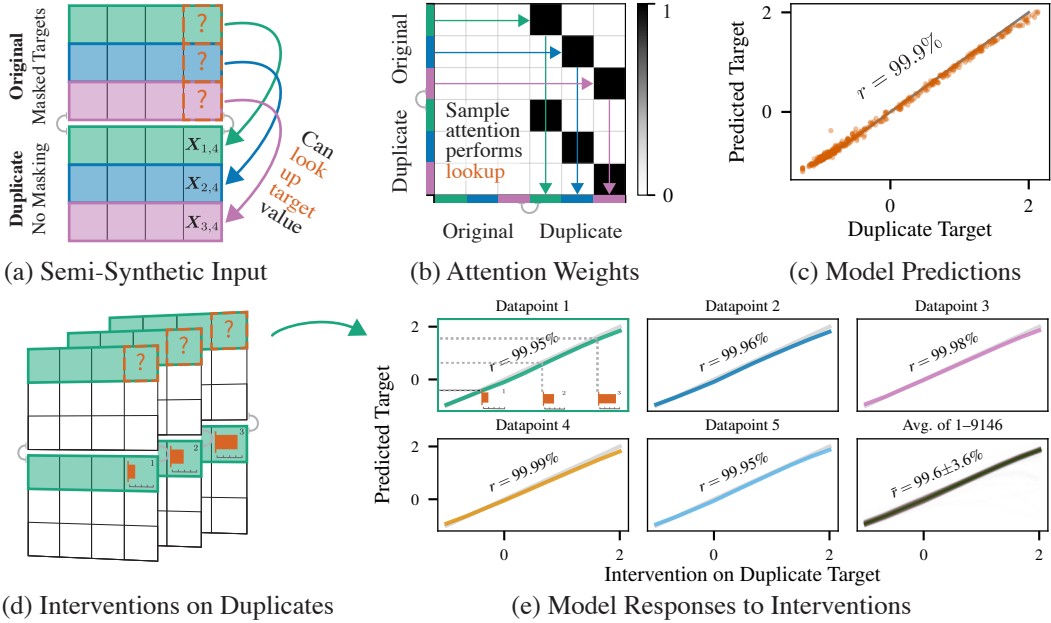

(a) Semi-Synthetic Input     (b) Attention Weights     (c) Model Predictions

(d) Interventions on Duplicates     (e) Model Responses to Interventions

Figure 3: Demonstrating NPT's ability to predict from Attention Between Datapoints (ABD). (a) We append to the original data with masked targets [?] a copy of the same data with all masked values revealed, such that perfect prediction via lookup is possible. (b) Attention weights indicate that the ideal lookup behavior is learned by NPT. Shown are actual values learned by NPT at head $0$ and depth $4$ for the first 3 datapoints. (c) NPT predictions closely match the ideal values. (d) Additionally, we intervene on the values of individual targets, (e) finding that NPT predictions adjust accordingly.

## 4.2 NPTs Can Learn to Predict Using Attention Between Datapoints

To determine if NPTs can successfully learn to exploit interactions between datapoints (**Q2**), we introduce a task with strong input correlations for which we know ground-truth interactions. Concretely, we use the UCI Protein regression dataset (cf. §4.1) to construct the following semi-synthetic task: for each batch, we input the original data with masked target values as well as a *copy* of the original data where all target values have been revealed, i.e., no masking is applied (Fig. 3a). NPTs can use attention between datapoints to achieve arbitrarily good performance by *learning* to look up the target values in the matching duplicate row. At test time, we input novel semi-synthetic test data to ensure that NPT has learned the correct relational mechanism and not just memorized target values.

NPTs successfully learn to perform this lookup between original and duplicate datapoints. The ABD attention weights, visualized for the first three datapoints in Fig. 3b, clearly show the model correctly attending to the duplicates. As a result, NPT predictions are Pearson-correlated with the duplicate targets at $r = 99.9\%$ (Fig. 3c). This equals an RMSE of only $0.44$, about a magnitude lower than the error on the original Protein dataset (Table 11). We conclude that NPTs learn to predict by looking up the target values from matching points. Further discussion and attention maps are in Appendix B.1.1.

Purely parametric models cannot exploit information from other datapoints, limiting their performance. For example, MLPs achieve an RMSE of 3.62 on this task. Non-parametric approaches also cannot solve this task in its original form, because unlike NPTs they must be told which datapoints are the originals (training data) and which the duplicates (test data) as well as which columns contain features and which target values. We demonstrate in Appendix B.1.2 that even when we make these concessions, we can easily adapt the task such that both k-Nearest Neighbors and Deep Kernel Learning fail to solve it. In fact, we are not aware of any other model that can solve the adapted task.

Additionally, we perform an *interventional* experiment to investigate the extent to which NPTs have actually learned the causal mechanism underlying the lookup task. As illustrated in Fig. 3d, we now intervene on individual duplicate datapoints at test time by varying their target value across a wide range. We stress that we perform these experiments without retraining the model, using exactly the same NPT from Figs. 3a-c. The model is now confronted with target values associated with features

Table 2: Drop in NPT performance after destroying information from other datapoints. Shown are changes in test set performance, where negative values indicate worse performance after corruption.

| $\Delta$ *Accuracy* | CIFAR-10 | Poker | Income | Higgs | MNIST | Forest | Kick | Breast Cancer |
|---|---|---|---|---|---|---|---|---|
| | −1.2 | −1.1 | −1.1 | −0.5 | −0.4 | −0.1 | −0.1 | 0.0 |

| $\Delta_{RMSE}/_{RMSE}$ (%) | Yacht | Protein | Boston | Concrete |
|---|---|---|---|---|
| | −52% | −21% | −20% | −7% |

that are highly unlikely under the training data. This label distribution shift [35] is a challenging setting for neural networks. However, NPT predictions follow the intervened target values with near-perfect correlation, Fig. 3e, continuing to predict by correctly looking up targets.

We now confidently conclude that NPTs robustly learn the causal data-generating mechanism underlying the semi-synthetic dataset. This requires NPTs to *learn* a non-trivial sequence of compuational steps. They must learn to match rows based on similarity of relevant features; to look up the target value of the duplicated datapoint; and, to copy that value into the target of the masked datapoint.

## 4.3 NPTs Learn to Use Attention Between Datapoints on Real Data

We next consider (**Q3**): do NPTs actually learn to use attention between datapoints for prediction on real data? We design a test that allows us to quantify the extent to which the predictions of an NPT trained in standard fashion on one of our benchmark datasets depend on relationships between datapoints at test time. Concretely, for each target value in the input we randomize the data for all *other* datapoints by independently shuffling each of their attributes across the rows. We then evaluate the loss on the prediction at the target entry and repeat this procedure for all test datapoints. This completely corrupts the information from all datapoints except the one for which we evaluate. Hence, a model that relies meaningfully on attention between datapoints will show deteriorating performance. We give an algorithm for the corruption procedure as well as further discussion in Appendix B.2.1.

We report the resulting change in performance after corruption in Table 2 for all datasets from §4.1. We find that for most datasets, the corruption of other rows at test time significantly decreases the performance of the trained NPT models. This indicates that the NPTs have successfully learned to make predictions supported by attention between datapoints. For some datasets, the corruption experiment deteriorates performance completely. For example, for the Protein regression dataset NPT achieves state-of-the-art performance, but corrupting the input at test time leads to NPT performing worse than all of the baselines considered in §4.1. We note that minor differences in performance are often still significant, as differences between competing models in §4.1 are often likewise small.

Interestingly, on certain datasets such as Forest Cover, Kick, and Breast Cancer, corrupted inputs do not significantly affect performance. It appears that when NPTs do not find it advantageous to rely on attention between datapoints during training, they can learn to completely ignore other inputs, essentially collapsing into a standard parametric model. This supports our earlier claims that NPTs can learn end-to-end from data the extent to which they rely on other datapoints for prediction. We think this is extremely interesting behavior and are unaware of prior work reporting similar results. However, we stress that these results reflect inductive biases of the NPT architecture and do not lend themselves to general statements about the performance of parametric versus non-parametric models.

## 4.4 NPTs Rely on Similar Datapoints for Predictions on Real Data

So far, we have presented convincing evidence that NPTs (sometimes strongly) depend on attention between datapoints. However, we do not know what kind of interactions are learned in practice on real data (**Q4**). As an initial step towards understanding this, we now present two experiments investigating *to which* other datapoints NPT attends.

**Qualitative Evidence.** Figure 4 shows an attention map for attention between datapoints (ABD) of NPT on a batch of the Protein regression dataset. We sort the input data with respect to their input space distance such that similar datapoints are now close to each other. The diagonal pattern

in Fig. 4 indicates that NPT attends more strongly to datapoints that are similar in feature space. Appendix B.3.1 discusses this further and gives additional attention maps.

**Quantitative Evidence.** Seeking a quantitative measure for this hypothesis, the *data deletion* experiment repeats the following procedure for all test set points: iteratively delete other datapoints from the input if they do not significantly affect the prediction. We stop if less than 2% of the original datapoints remain, or if the total change in prediction for the target (relative to the original prediction with all data) exceeds 10%. We investigate the average input feature space distances between the test point and the *kept* datapoints, as well as the distances between the test point and the *deleted* datapoints. "Input features" here refer to all attributes of the input datapoints that are not labels.

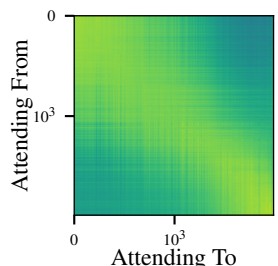

Fig. 4: Attention weights.

We find that kept datapoints have a significantly lower average feature space distance to the test point than those deleted. This indicates that two datapoints $i, i'$ that are similar in input feature space, such that $\sum_{j<d} (X_{i,j} - X_{i',j})^2$ is low, have a larger effect on the predictions of one another. A Wilcoxon signed-rank test is significant at $p \approx 8.77 \cdot 10^{-130}$. We give full details on this in Appendix B.3.2.

Both experiments support the hypothesis that NPTs rely on similar datapoints for prediction in real data settings. One possible explanation is that similar datapoints might have different realizations of observation noise which NPTs could learn to average out. Altogether, we conclude that NPTs can and do learn representations which rely on interactions between datapoints for prediction.

## 5 Limitations, Future Work, and Conclusions

**Limitations.** NPTs share scaling limitations with all naïvely non-parametric approaches [74] and GNNs [52]. We demonstrate this in a preliminary analysis of the computational cost of NPTs and the baseline methods – including training time and CPU/GPU memory requirements – in Appendix B.6. While we have seen success with random minibatching (§2.6), future work might consider applying principled attention approximations, such as learning representative input points [59], kernelization [19, 47], or other sparsity-inducing methods [5, 18, 84], to improve the scalability of NPTs.

**Future Work.** We believe that the unique predictive mechanism of NPTs makes them an interesting object of study for other tasks including continual learning, multi-task learning, few-shot generalization, and domain adaptation. For example, when predicting under distribution shift, general relations between datapoints and attributes may remain valid and allow NPTs to accommodate such scenarios better. Additionally, future work could explore the connections to stochastic processes, e.g., by extending NPTs to be approximately consistent, similar to Neural Processes [36, 37, 49].

**Conclusions.** We have introduced Non-Parametric Transformers (NPTs), a novel deep learning architecture that takes the entire dataset as input and uses self-attention to model complex relationships *between* datapoints. NPTs challenge and naturally extend parametric modeling as the dominant paradigm of deep learning. They have the additional flexibility to learn to predict by directly attending to other datapoints. Notably, NPTs learn this end-to-end from the data at hand. Empirically, NPTs achieve highly competitive performance on a variety of benchmarks, and additional experiments demonstrate their ability to solve complex reasoning tasks over datapoints. Further, we show that on real data, NPTs learn to rely on attention between datapoints for prediction. We believe that the characteristics of NPTs will make them an exciting object of further study.

## Acknowledgments and Disclosure of Funding

We acknowledge funding from the New College Yeotown Scholarship (JK), the Rhodes Trust (NB), and the Open Philanthropy AI Fellowship (CL). We thank Lewis Smith, Pascal Notin, Uri Shalit, Joost van Amersfoort, Sören Mindermann, Lood van Niekerk, and the anonymous reviewers for helpful feedback and interesting discussions that have led to numerous improvements of the paper.

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
