# Self-Attention Between Datapoints: Going Beyond Individual Input-Output Pairs in Deep Learning

# Appendix

## Table of Contents

# A Proof – NPT Is Equivariant over Datapoints

We here provide proof that NPT is equivariant to a permutation of the datapoints. This requires, among other things, showing that multi-head self-attention is equivariant. We were unable to find this proof in the existing literature, e.g., Set Transformer [59] relies heavily on equivariance of self-attention but does not provide proof. In the following, we will refer to datapoints as the *rows* of our input, see e.g., Fig. 1.

**Definition 1.** *A function $f : \mathcal{X}^n \rightarrow \mathcal{X}^n$ is row-equivariant if for any permutation $\sigma : [1, \ldots, n] \rightarrow [1, \ldots, n]$ applied to the dimensions of $\mathcal{X}^n$, we have for all $i$, $f(X_1, \ldots, X_n)[i] = f(X_{\sigma^{-1}(1)}, \ldots, X_{\sigma^{-1}(n)})[\sigma(i)]$.*

**Lemma 1.** *Any function of the form $f(X_1, \ldots, X_n) = (g(X_1), \ldots, g(X_n))$ for some $g$ is row-equivariant. These functions are denoted as 'row-wise operations', as they consist of the same function applied to each of the rows of the input.*

*Proof.* Follows immediately from the structure of $f$. $\qquad\square$

**Lemma 2.** *The composition of row-equivariant functions is row-equivariant.*

*Proof.* This result is widely known, but a proof here is included for completeness. Let $f$ and $g$ be row-equivariant.

$$f \circ g(\sigma X) = f(g(\sigma X)) = f(\sigma g(X)) = \sigma f(g(X)). \tag{8}$$

$\qquad\square$

**Lemma 3.** *Let $W \in \mathbb{R}^{n \times m_1}$ and $X \in \mathbb{R}^{m_2 \times n}$. The function $X \mapsto XW$ is row-equivariant.*

*Proof.* Let $\sigma X$ be a permutation of the rows of $X$. Then we have

$$(\sigma X)W[i,j] = \sum \sigma X[i,k]W[k,j] \tag{9}$$

$$= \sum X[\sigma^{-1}(i), k]W[k,j] = XW[\sigma^{-1}(i), j] = \sigma(XW)[i,j]. \tag{10}$$

$\qquad\square$

**Lemma 4.** *The function $X \mapsto Att(XW^Q, XW^K, XW^V)$ is row-equivariant.*

*Proof.* Let the row-wise softmax function be denoted $\omega(\cdot)$. Then we have

$$Att(XW^Q, XW^K, XW^V) = \omega(XW^Q(XW^K)^\top / \sqrt{h})XW^V, \tag{11}$$

where

$$\sigma XW^Q(\sigma XW^K)^\top[i,j] = \sigma(XW^Q)\sigma(XW^K)^\top[i,j] \tag{12}$$

$$= \sum \sigma(XW^Q)[i,k]\sigma(XW^K)[j,k] \tag{13}$$

$$= \sum XW^Q[\sigma^{-1}(i), k]XW^K[\sigma^{-1}(j), k] \tag{14}$$

$$= XW^Q(XW^K)^\top[\sigma^{-1}(i), \sigma^{-1}(j)] \tag{15}$$

$$=: A. \tag{16}$$

Note that the above result states that the function $XW^Q(XW^K)^\top$ is *not* row-equivariant because of the additional permutation of the columns. Let $\sigma$ denote a permutation operator on matrices. Then straightforwardly we have the following:

$$\omega(\sigma A / \sqrt{h})) = \sigma\omega(A / \sqrt{h}). \tag{17}$$

Finally, it remains to show that the final matrix multiplication step restores the row-equivariance property we seek.

$$\sigma \underbrace{\omega(XW^Q(XW^K)^\top/\sqrt{h})}_{=:M}(\sigma XW^V)[i,j] = \sigma(M)(\sigma XW^V)[i,j] \tag{18}$$

$$= \sigma(M)\sigma(XW^V)[i,j] \tag{19}$$

$$= \sum M[\sigma^{-1}(i), \sigma^{-1}(k)](XW^V)[\sigma^{-1}(k), j] \tag{20}$$

$$= M(XW^V)[\sigma^{-1}(i), j]. \tag{21}$$

Which shows that self-attention is row-equivariant. □

**Lemma 5.** *The following hold:*

1. *Multihead self-attention is equivariant.*
2. *If $f$ and $g$ are row-equivariant, then the function $x \mapsto g(x) + f(x)$ is also row-equivariant.*
3. *Res(H) is row-equivariant.*
4. *MHSA(H) is row-equivariant.*
5. *ABD is row-equivariant.*
6. *ABA is row-equivariant.*

*Proof.* We show each item.

1. We know that $X \mapsto O_i$ is equivariant from the previous lemma, and this trivially implies that $X \mapsto \mathrm{concat}(O_1, \ldots, O_k)$ will also be row-equivariant. Finally, because $\sigma AB = \sigma(AB)$, get that MHSelfAtt(H) is row-equivariant.
2. Straightforward.
3. Because LayerNorm is row-equivariant (being a function applied row-wise to the matrix), Res(H) is a sum of two row-equivariant functions and so by a previous result will also be row-equivariant.
4. Because rFF is again a row-wise operation and so trivially row-equivariant, the previous results on sums and compositions of row-equivariant functions directly yield row-equivariance of MHSA.
5. ABD is by definition an application of MHSA(H), and therefore is row-equivariant by the above result.
6. ABA is a row-wise operation and is therefore trivially row-equivariant.

□

**Property A.0.1.** *NPT is row-equivariant.*

*Proof.* Each layer of NPT has been shown to be row-equivariant. Because NPT is a composition of such row-equivariant functions, it is therefore row-equivariant. □

# B  Additional Results

## B.1  Semi-Synthetic Experiments

### B.1.1  Attention Maps for the Semi-Synthetic Experiments

We here display additional results for the semi-synthetic experiments of Section 4.2. In Fig. B.1, we display attention weights for Attention Between Datapoints (ABD) for all depths and a subset of heads of the architecture. We see that some, but not all, attention heads display the desired diagonal lookup pattern. Note that, in this case, one head would suffice to implement lookup and perfectly solve the task.

A brief comment on the attention maps with the "double diagonal" structure (e.g., depth 4, head 0): we see that (a) original datapoints attend to the duplicate points and (b) duplicates also attend to duplicate datapoints. Behavior (a) makes sense: NPT needs to attend to the duplicates from the originals to look up the target values. This behavior in turn minimizes loss. Behavior (b) is irrelevant to loss, because NPT does not need to predict anything for the duplicates, and no loss is computed. However, (b) suggests that the query embeddings learned by the self-attention *ignore* the masked

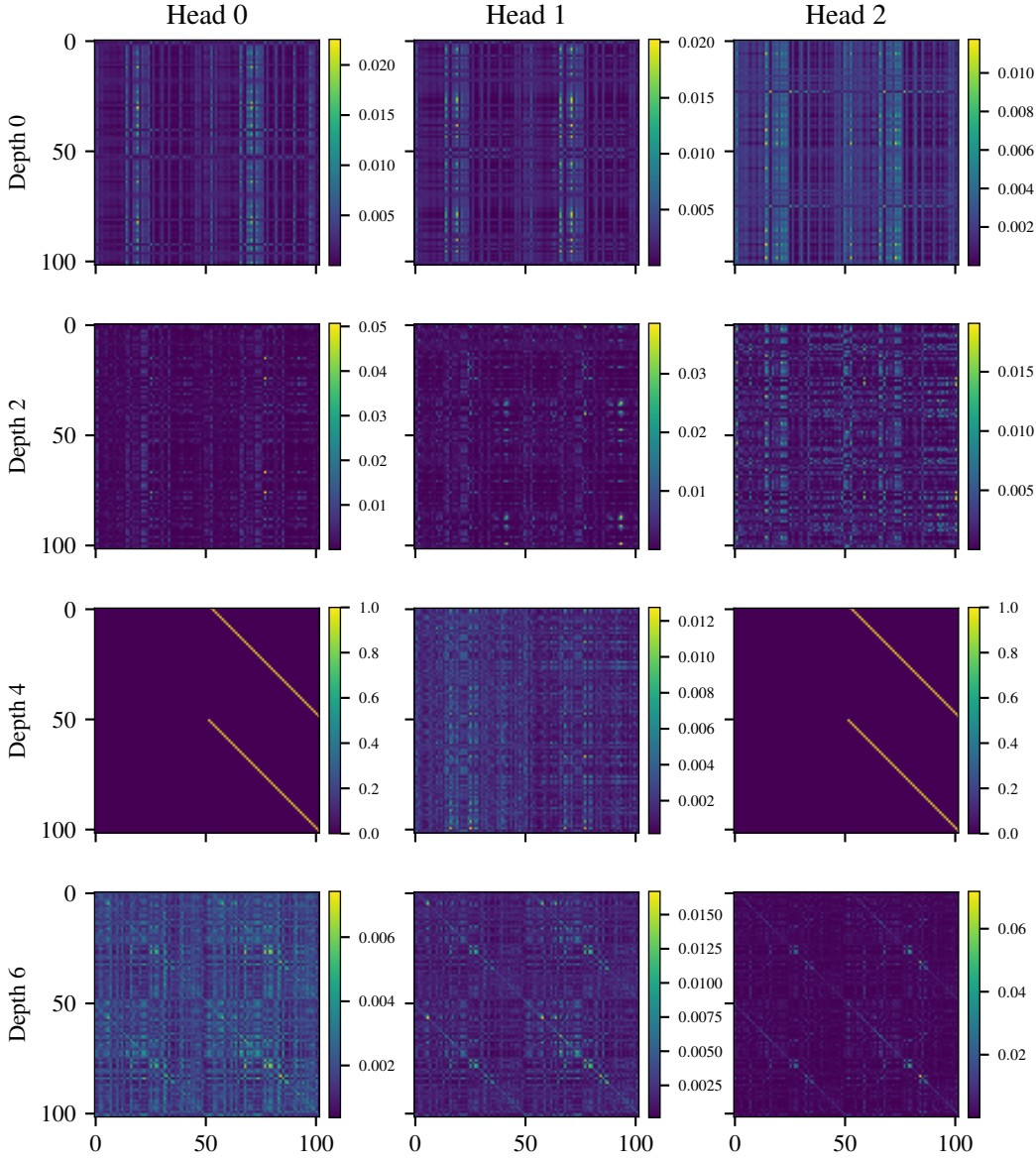

Figure B.1: Visualizations of NPT attention maps for Attention Between Datapoints (ABD) for the semi-synthetic experiment at all model depths, a selection of heads, and a single batch of input data. Evidently, not all attention maps need to perform a "lookup" for the model to solve the task. In fact, some heads appear to learn almost query-independent behavior (e.g., heads 0, 1, and 2 at depth 0).

out label column in the input. Hence, the resulting queries for the originals and the duplicates would be identical – both leading to high attention values for the keys of the duplicates – and ultimately resulting in the double diagonals in Fig. B.1.

## B.1.2 Modified Semi-Synthetic Experiments

**Setup.** In Section 4.2, we mention that with some concessions the original lookup task can also be solved by standard non-parametric models. However, we also mention that simple modifications to the task make it, again, unsolvable for any model of which we are aware other than NPT. We here demonstrate these hypotheses for two non-parametric models: k-Nearest Neighbors (k-NN) and Deep Kernel Learning (DKL).

Table 3: Variations of the semi-synthetic dataset that require learning of between-datapoint interactions more complex than simple lookups. While NPTs can learn complex interactions between datapoints, conventional non-parametric approaches lack flexibility and fail.

| Test RMSE ↓ | Original Synthetic | Random Feats. | Add One | Random Feats. + Add One |
|---|---|---|---|---|
| 1-NN | **0.00** | 7.19 | 6.11 | 7.80 |
| k-NN | **0.00** | 5.42 | 5.18 | 5.64 |
| DKL | **0.00** | 5.94 | 6.31 | 6.36 |
| NPT | 0.34 | **0.24** | **0.46** | **0.75** |

First, we apply k-NN and DKL to the original duplication tasks. As mentioned in the main text, this already requires us to make some concessions: we now need to explicitly split the input data into a global training set (all duplicated datapoints) as well as a test set (all original datapoints). That is, if all duplicate datapoints make up the training set, then non-parametric models are able to predict perfectly on the original datapoints, because most non-parametric models rely on distances in some manner, and here, distances in input feature space are sufficient to successfully match entries. This is trivially true for k-NN but also for DKL, where the RBF kernel of the GP will lead to the desired "matching behavior" as long as the learned neural network embedding does not collapse distances.

In other words, NPTs would ideally learn a k-NN-style prediction for the semi-synthetic dataset. Crucially, while non-parametric models predict based on distances because of fixed design choices, NPTs *learn* this behavior and can just as well learn other more complicated relations between datapoints.

We now present two modifications to the semi-synthetic dataset; NPT can accommodate them because the model learns the nature of interactions, but they significantly affect the performance of the fixed kernel methods.

- **Random Features**: A subset of the features are randomized across both original and duplicate datapoints independently. Specifically, we overwrite the entries of the last three features with noise drawn independently from a Gaussian distribution $\mathcal{N}(1, 1)$. To solve the task, matches between datapoints must now be computed using the subset of non-randomized features only.
- **Add One**: We add 1 to all target regression values *only* for the duplicate datapoints. Matches can still be made based on all features, but now a 1 must be subtracted from the lookup value to solve the task.

As in the original setting, we train the models on the modified semi-synthetic datasets and check with novel test data whether they have learnt the correct relational mechanism underlying the experiment.

Note that the Random Features and Add One settings also distinguish our setup from prompting in natural language processing literature [12, 72] because the original datapoints are no longer "correct" input-output pairs; the model must use an underlying relational structure instead of memorization to solve the task.

**Results.** Table 3 presents RMSE values obtained by the models when trained on the original duplication task, the two modifications separately, as well as both modifications applied.

Evidently, for NPTs, the different scenarios do not lead to a large difference in performance; in all instances, they achieve near-perfect loss because their predictions leverage attention between datapoints. Careful optimization of NPT training convergence would likely lead to a further reduction in loss. Nevertheless, the achieved losses by NPT are more than a magnitude lower than those on the original data and correspond to a near-perfect Pearson-correlation with the target values of $r > 99.9\%$. We conclude that NPTs successfully learn to attend to the correct subset of features, to subtract 1 from the lookup target values, or to do both at the same time.

Next, we consider the non-parametric models. First, we confirm in *Original Synthetic* that the non-parametric models can indeed solve the original lookup task. However, we find that neither DKL

nor k-NN can accommodate any of the modifications, reverting to an RMSE that is worse than the performance of all baselines on the original Protein dataset, see Table 11.[5]

For $k$-Nearest Neighbor, $k = 1$ is clearly optimal in the original semi-synthetic setup. However, k-NN cannot learn to ignore certain attributes (Random Features) and or to modify looked-up values. Setting $k > 1$ actually improves prediction because it considers other matching points in addition to the (now misleading) duplicates for prediction. However, even with $k > 1$, k-NN does not achieve much better than guessing performance on the modified tasks.

DKL also fails to accommodate any of the presented task modifications. We suspect that DKL, in theory, should be able to solve the Random Features task. That is, DKL should be able to use the neural network to learn a representation that discards any information from the randomized columns. We were unable to achieve this, but it may be possible with additional adaptations to the model. Ideally, we would condition the GP on new "test data" (the duplicates) in each minibatch during training. This was not easily possible with the GPyTorch codebase.[6] At test time however, we did directly reconstruct an exact GP using embedded inputs and RBF scale parameters learned during training.

In any case, DKL can never solve the Add One scenario because, after independently transforming features with a neural network, DKL simply applies a GP in embedding space. This means that it will always naively interpolate target values between training data (duplicates) and test data (features) in embedding space, and cannot *learn* interactions between points, such as subtracting 1 from all duplicate targets.

Even further, there is another easy option of how to construct this experiment such that only NPT will be able to solve it: we could *randomly sample the attribute* for which we mask out the entry, i.e., all columns can now be target columns. All non-parametric models presented here rely on a fixed set of features as input to predict for a fixed target column. They are not compatible with this style of "imputation" problem, i.e., there is no way to even take as input data like this in such models. NPTs, however, take both features and targets as input, only using the masking mechanism to distinguish between features and targets as well as train and test data. Hence, they can easily adapt to this scenario.

The bad results for the non-parametric models also highlight that these models must predict non-parametrically, unlike NPT, which could always fall back to parametric prediction if it cannot learn the interactions required for a task.

**(k)-NN Hyperparameter details.** We use the scikit-learn [69] implementation of (k)-Nearest Neighbors, where we exhaustively search for neighbors by setting `algorithm=brute` and otherwise use default parameters. For 1-NN, we set $k = 1$, for $k$-NN we sweep over $k \in [1, \dots, 10]$ and report results for the $k$ that achieved the best performance.

**DKL Hyperparameter details.** We use the GPyTorch implementation of Deep Kernel Learning. We perform a non-exhaustive random sweep over a selection of hyperparameters and select those with best validation performance. This results in the following changes from the default hyperparameter values: for the Original Synthetic and Add One scenario we disable dropout, use hidden layers $[100, 100]$, a learning rate of $0.0001$, train for a maximum of $30000$ epochs, with $256$ inducing points, $8$ features, batch size of $128$, and early stopping patience on the validation loss of $20$ epochs. For the Random Features and the Random Features + Add One scenarios, we arrive at the same configuration, except that we train with $64$ inducing points.

---

[5]In fact, the RMSEs are about equal to the standard deviations of the target values in the Protein dataset, $6.11$, such that the values obtained by the models on the modified setups amount to random guessing. We further note that we apply all modifications to the standardized input data, such that the Add One setting adds a full standard deviation for the final evaluation in Table 3.

[6]Gardner, Jacob R., et al. "Gpytorch: Blackbox matrix-matrix gaussian process inference with gpu acceleration." NeurIPS 2018.

## B.2 Attention Between Datapoints on Real Data

### B.2.1 Corruption Experiments

In our Data Corruption experiments in Section 4.3, we make use of Algorithm 1 below. When predicting for a datapoint $k$, this algorithm completely destroys information from all other datapoints $i \neq k$ in the batch $b$ by randomly permuting attribute values across all other datapoints. Therefore, if NPT's loss increases after corruption, it must meaningfully rely on attention between datapoints for prediction.

---

**Algorithm 1:** Data Corruption

---

**Input:** list of masked minibatches $\mathcal{B} = [\boldsymbol{X}^{(b)} \in \mathbb{R}^{K \times d} \mid b \in 1 \ldots B]$, unmasked label column
   $\boldsymbol{X}_{:,d}$, trained model $f : \boldsymbol{X}^{(b)} \to \boldsymbol{X}^{(b)}$, batch size $K$, loss function $\mathcal{L}$, number of
   attributes (including features and target) $d$

**Returns:** test loss under data corruption $\mathcal{L}^{\text{corr}}$

$\mathcal{L}^{\text{corr}} \leftarrow 0$
**for** $\boldsymbol{X}^{(b)}$ *in* $\mathcal{B}$ **do**
  **for** $k$ *in* $1 \ldots K$ **do**
    $\boldsymbol{X}^{(b,k)} \leftarrow \boldsymbol{X}^{(b)}$        // initialize batch to be corrupted
    **for** $j$ *in* $1 \ldots d$ **do**
      $\boldsymbol{X}^{(b,k)}_{i \neq k, j} \leftarrow \texttt{permute}_{\text{axis}=i}(\boldsymbol{X}^{(b,k)}_{i \neq k, j})$  // permute each attr. column indep.
    **end**
    $\mathcal{L}^{\text{corr}} \mathrel{+}= \mathcal{L}(f(\boldsymbol{X}^{(b,k)})_{k,d}, \boldsymbol{X}_{k,d})$ // compute loss w/ unmasked label column
  **end**
**end**
**return** $\mathcal{L}^{\text{corr}}$

---

Alternatively, we could also input datapoints *individually*, i.e., decrease the minibatch size to 1, to test if NPT depends on attention between datapoints. Indeed, we find that performance also deteriorates in this scenario. However, we believe that the Data Corruption experiment provides stronger evidence because it preserves batch statistics across attributes. This makes sure that performance deterioration is not caused by spurious factors, such as a decreased batch size that was not encountered in training. While NPT is generally compatible with varying batch sizes, we leave a thorough investigation of this for future work.

## B.3 Real Data – *To Which* Other Points Does NPT Attend?

### B.3.1 Attention Maps on Real Data

In Fig. B.2, we display ABD attention maps of NPT for the Protein regression dataset in addition to the one shown in Section 4.4. For visualization purposes, we sort the input datapoints with respect to their feature space distance to an arbitrary test datapoint. This is to ensure that the global structure of the attention maps in Fig. B.2 has meaning. Specifically, nearby entries in the attention maps belong to input datapoints that are close in input space. With this transformation, the diagonal patterns appearing in Fig. B.2 clearly suggest that our model is attending more strongly between datapoints that are similar in input space. Similar to the semi-synthetic experiments, some but not all attention heads display this pattern of interest.

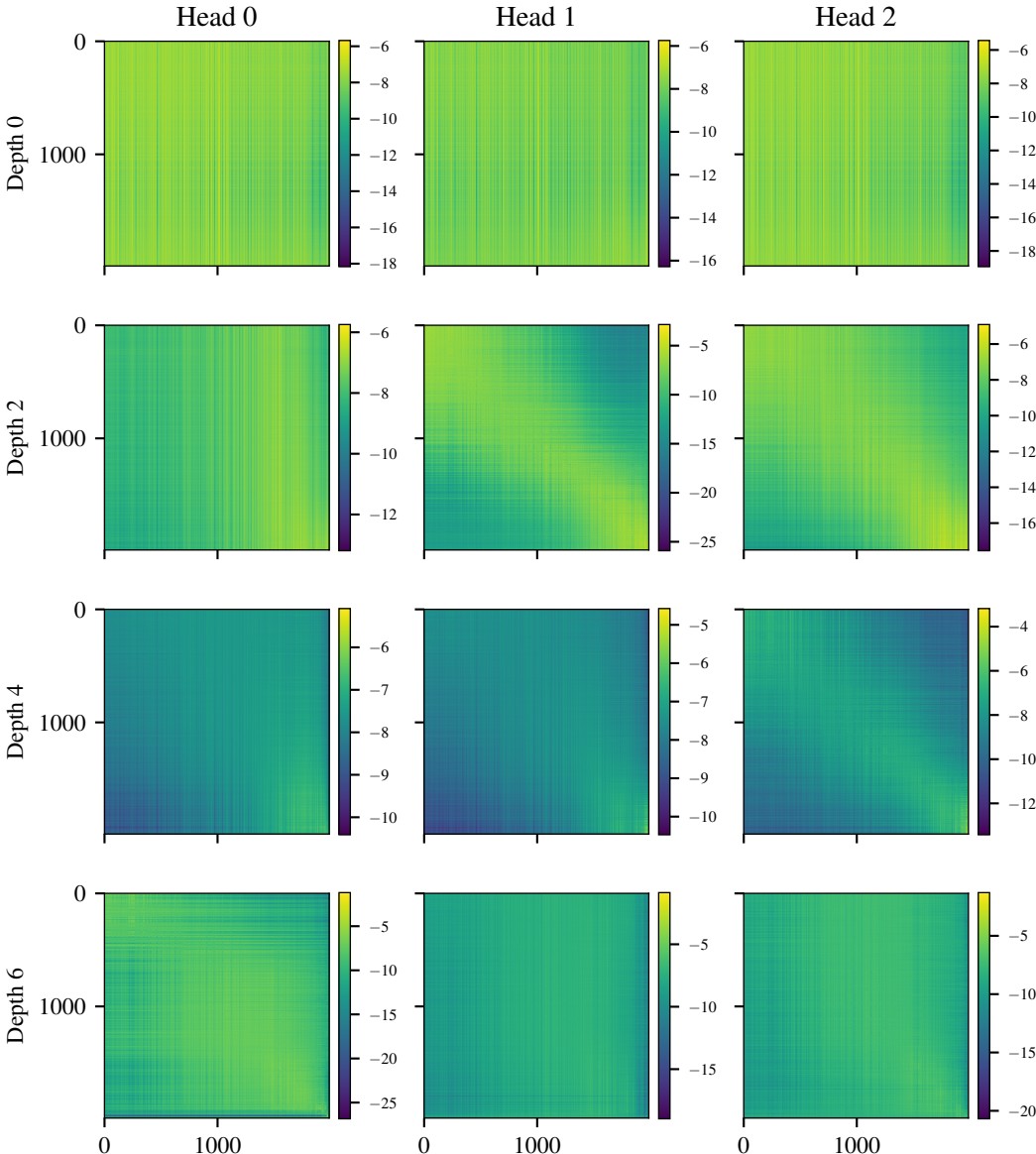

Figure B.2: Visualizations of the Attention Between Datapoints (ABD) attention maps for real data – here, the Protein regression dataset – for all depths and a selection of heads. Input to the model is sorted such that datapoints that are similar in input space have nearby indices. The diagonal pattern (e.g., depth 2 and head 1) indicates that the model attends to similar inputs more strongly. For illustration purposes, we here plot the log of the attention values.

---
**Algorithm 2:** Data Deletion
---
1 **Input:** Masked data $\boldsymbol{X} \in \mathbb{R}^{n \times d}$, active sample index $i^*$.

2 $\hat{y} \leftarrow \text{NPT}(\boldsymbol{X})_{i^*,d}$            `// original NPT prediction at active datapoint`
3 $\Delta_{\max} \leftarrow 0.1$              `// maximum allowed change in prediction`
4 $\Delta_{\text{it}} \leftarrow 0.01$        `// initialize maximum change per deleted datapoint`
5 $N_{\text{max-retry}} \leftarrow 50$       `// maximum number of retries before increasing` $\Delta_{\text{it}}$
6 $\epsilon \leftarrow 0.02$         `// fraction of points remaining at which we break`
7 $\mathcal{R} \leftarrow \{1, \ldots, n\} \setminus \{i^*\}$             `// initialize remaining set`
8 $N_{\text{retry}} \leftarrow 0$             `// initialize no. of retries`

9 **while** *True* **do**
10    $c = \texttt{random\_choice}(R)$         `// random proposal for data deletion`
11    $\hat{y}_{\text{proposal}} = \text{NPT}(\boldsymbol{X}_{(\mathcal{R} \setminus \{c\}) \cup \{i^*\}})_{i^*,d}$     `// predict without proposed datapoint`
12    $\Delta_{\text{proposal}} = \frac{|\hat{y}_{\text{proposal}} - \hat{y}|}{\hat{y}}$       `// change in pred. when deleting proposal`
13    **if** $\Delta_{\text{proposal}} < \Delta_{it}$ **then**
14      **if** $\Delta_{\text{proposal}} < \Delta_{max}$ **then**
15        $\mathcal{R} \leftarrow \mathcal{R} \setminus \{c\}$          `// delete datapoint from input`
16        $N_{\text{retry}} \leftarrow 0$
17      **else**
18        break               `// exceeded maximum change`
19    **else**
20      $N_{\text{retry}} \leftarrow N_{\text{retry}} + 1$      `// candidate change was too large, try again`
21    **if** $N_{\text{retry}} \geq N_{\text{max-retry}}$ **then**
22      $\Delta_{\text{it}} \leftarrow 1.1 \cdot \Delta_{\text{it}}$        `// increase allowed change per iteration`
23      $N_{\text{retry}} \leftarrow 0$
24    **if** $|\mathcal{R}| < \epsilon \cdot n$ **then**
25      break       `// less than` $\epsilon\%$ `of original datapoints remaining`
**end**
26 **return** $\mathcal{R}$

---

### B.3.2 Data Deletion Experiment

We here give full details on the Data Deletion experiment presented in Section 4.4. To recap, we consider the prediction of NPT for a single test sample $i^*$. We then iteratively delete other datapoints from the input if they do not significantly change the prediction of NPT on $i^*$. Algorithm 2 describes this in detail. We are then interested in differences between the deleted and the kept datapoints. Specifically, we compare the average feature space distance in input space between the active datapoint $i^*$ and either the kept datapoints $\mathcal{R}$ or deleted datapoints $\{1, \ldots, n\} \setminus (\{i^*\} \cup \mathcal{R})$, obtaining average distances $D_{i^*, \text{kept}}, D_{i^*, \text{deleted}}$. We break out of the deletion algorithm if less than $\epsilon\%$ of the original points remain, to reduce variance in our estimates of the kept statistic. We repeat Algorithm 2 for all 5567 test points $i^* \in \mathcal{D}_{\text{test}}$ in the Protein regression dataset.

We perform a Wilcoxon signed-rank test on the pairs $\{D_{i^*, \text{kept}}, D_{i^*, \text{deleted}}\}_{i^* \in \mathcal{D}_{\text{test}}}$ to determine if the median of the kept datapoints is less than the median of the deleted ones. The test is highly significant at $p \approx 0$, i.e., smaller than the floating point precision of SciPy Stats allows. The raw Wilcoxon statistic is 3125889.5.

To make sure the difference is not an effect of sample size, we also construct a set of average differences to a set of randomly drawn datapoints.[7] That is, instead of using Algorithm 2 for *targeted* deletion, we *randomly* construct $\mathcal{R}$, essentially only applying lines 10 and 15 of Algorithm 2. For

---
[7] There are many fewer kept than deleted datapoints. Further, there are outliers in the dataset, and these affect the deleted datapoints more often than the kept datapoints. We find that the average distance between a *random* subset and the *deleted* (not the kept!) datapoints also becomes statistically significantly smaller at large sample sizes. Hence, we compare the *deleted* datapoints to a *random* subset to control for size effects.

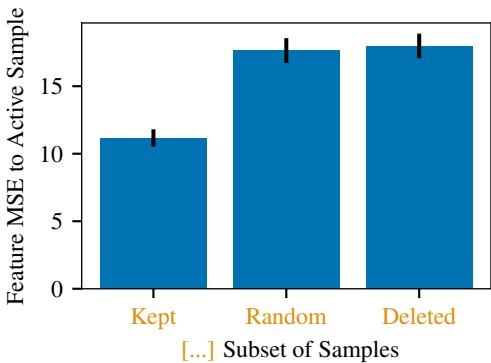

Figure B.3: When predicting for any given datapoint, NPT prefers to keep similar datapoints around. Displayed are average feature space differences and their standard errors between the active datapoint and the sets of kept, random, and deleted datapoints for a single batch.

each active test row $i^*$, we randomly delete as many datapoints as were deleted in targeted fashion. A Wilcoxon signed-rank test between the distances for the random and kept subset is likewise significant at $p \approx 8.77 \cdot 10^{-130}$. This is the value we report in the main body.

We also run a computationally more demanding version of the algorithm with $\Delta_{it} \leftarrow 0.005, \epsilon \leftarrow 0.01$ to see how many points we can successfully delete. This version of the algorithm requires more computation which is why we limit execution to the test datapoints of a single batch. The results are statistically significant at $5.26 \cdot 10^{-49}$ for kept < deleted and $8.38 \cdot 10^{-39}$ for kept < random for a Wilcoxon signed-rank test. We illustrate the differences between the distances in Fig. B.3. We further note that using Algorithm 2, we are able to reduce the set of datapoints present in the input to 1% of the original $n$ for 79.5% of active test datapoints and to 10% in 99.5% of cases. Percentages refer to $n = 2048$ datapoints in total, of which 398 were test datapoints.

All in all, these experiments strongly suggest that NPT relies on interactions between similar datapoints for prediction.

### B.4   Ablation Study 1: NPT Hyperparameters

We conduct an ablation study on the Protein and Boston Housing datasets (Table 4). For Protein, the same 0.7/0.1/0.2 train/validation/test split is used for all model configurations. Boston Housing uses a 0.7/0.2/0.1 train/validation/test split with 10-fold cross-validation.

Despite the significant difference in dataset sizes between Boston Housing ($n = 506$) and Protein ($n = 45730$), and the fact that Boston Housing includes both categorical and continuous variables, the base models used for each dataset are nearly identical.

On both datasets, we use an NPT model with 8 layers, 8 heads, per-attribute hidden dimension $e = 128$, feature and target masking with $p = 0.15$ for each, a cosine annealing schedule for the loss tradeoff $\lambda$, the LAMB [98] optimizer with Lookahead [99], a flat-then-anneal learning rate schedule with cosine decay and base learning rate 0.001, dropout with rate 0.1 on the attention weights and after linear layers, and gradient clipping at 1. This configuration is essentially the same as the NPT-Base configuration described in Appendix C.1, which we use with minimal per-dataset modifications for all other results in this work.

Different in our base models between the two datasets are the following settings. The Boston Housing model takes as input the full dataset (i.e., batch size $= 507$) and Protein uses minibatching with batch size $= 2048$. Boston Housing trains for $20\,000$ steps, and Protein for $400\,000$. The learning rate is constant for the first $70\,\%$ of steps for Protein, but only for the first $50\,\%$ of steps for Boston, starting the learning rate annealing earlier to defend against overfitting on the small dataset. These changes directly result from the different dataset sizes.

As Table 4 shows, the performance of NPT is robust to a variety of significant hyperparameter choices. This illustrates that practitioners will likely *not need to spend much time tuning hyperparameters*

Table 4: NPT ablation study: test root mean-squared error (RMSE) on the Protein and Boston Housing regression datasets.

| Test RMSE ($\pm$ Std Err) $\downarrow$ | Protein | Boston |
|---|---|---|
| Base NPT | 3.41 | $3.00 \pm 0.23$ |
| No Semi-Supervision | 3.38 | $3.38 \pm 0.46$ |
| No Target Masking | 3.32 | $2.93 \pm 0.18$ |
| No Feature Masking | 3.56 | $2.95 \pm 0.21$ |
| No Feature Masking, No Target Masking | 3.58 | $3.20 \pm 0.26$ |
| Feature Mask $p = 0.15 \rightarrow p = 0.5$ | 3.87 | $3.39 \pm 0.23$ |
| Target Mask $p = 0.15 \rightarrow p = 0.5$ | 3.37 | $3.11 \pm 0.28$ |
| $8 \rightarrow 4$ Layers | 3.43 | $3.30 \pm 0.41$ |
| $8 \rightarrow 16$ Layers | 3.36 | $3.05 \pm 0.24$ |
| $8 \rightarrow 4$ Heads | 3.42 | $3.25 \pm 0.30$ |
| $8 \rightarrow 16$ Heads | 3.37 | $3.20 \pm 0.39$ |
| Tradeoff $\lambda = 0.5$ | 3.50 | $2.96 \pm 0.25$ |

when applying NPT to novel datasets. We now give results for the ablation study on the Protein and Boston datasets separately.

**Protein Dataset.** See Table 4 for results and performed ablations. It is computationally too expensive for us to perform full cross-validation over all ablations for the Protein regression dataset. Instead, we report the results of a single 5-fold cross-validation for the Base NPT configuration on Protein (also varying the model random state). This results in an RMSE of $3.40 \pm 0.05$ ($\sigma$). The standard deviation of the 5-fold cross-validation allows us to roughly gauge which ablations have significant effect. Given the results in Table 4, we find that the majority of ablations do not lead to meaningful changes in performance. Only the somewhat dramatic changes to the optimization of NPT result in its performance falling from the top rank on the Protein Dataset (second rank CatBoost has RMSE $= 3.51$): removing stochastic feature masking ($p_{\text{feature}} = 0$), removing both stochastic feature masking ($p_{\text{feature}} = 0$) and stochastic target masking ($p_{\text{target}} = 1$, training targets are always masked out at training time and NPT therefore cannot learn to attend to training targets at test time), or changing $p_{\text{feature}}$ to 0.5 (meaning that 50% of all input features are masked out). NPT appears to be particularly robust to changes in model complexity, e.g., depth and number of heads, although the results suggest that we could have further increased the size of Base NPT to achieve slightly higher performance.

**Boston Dataset.** See Table 4 for results and performed ablations. For the Boston dataset, we repeat ablations over all 10 CV splits. Similarly, ablations on the Boston dataset are largely inconsequential; none of them result in a statistically significant change in performance from the base model. The second rank performer on Boston is MLP, at RMSE $= 3.32$. Only ablation of semi-supervision or changing $p_{\text{feature}}$ to 0.5 result in a change in the top ranking of NPT among the baselines.

Altogether, the ablation study supports the claim that NPT can be applied successfully with very little tuning to datasets of vastly different sizes and feature types. Changes in model depth and number of heads do not appear significant, but using a reasonably low feature masking probability (e.g., 15%, as has been commonly used in the literature [24]) may be important to stable training.

Supported by these ablations, we sweep over only a small selection of configurations for our main benchmark comparison in Section 4.1. And indeed, it seems that NPT is robust to hyperparameter changes, given that these configurations perform well across vastly different settings (binary and multi-class classification, datasets with millions of datapoints, etc.) than those explored in the ablations. See Appendix E for details.

We speculate that NPT's robustness stems from (a) being a relatively overparametrized architecture that is powerful enough to model a wide variety of datasets and (b) from the effective regularization introduced by the feature masking mechanism. Finally, we emphasize that the aim of this work is to introduce the NPT architecture and examine its properties, not to spend significant effort and compute resources on achieving top performance across all benchmarks.

Table 5: Additional ablation studies. We study ablations of NPT (a) without ABA layers and (b) without stochastic feature masking. In both cases, performance tends to decrease. These results suggest that both ABA layers and stochastic feature masking contribute positively to the performance of NPTs. For the small datasets, we report mean values and standard errors over 10 CV splits.

| | NPT without ABA | NPT without Feature Masking | Default NPT |
|---|---|---|---|
| **Classification** | | | |
| Poker Hand (Acc. ↑) | 57.4 | 69.7 | **99.3** |
| Forest Cover (Acc. ↑) | 95.5 | 96.0 | **96.7** |
| Higgs Boson (AUC ↑) | 0.859 | 0.871 | **0.892** |
| Income (AUC ↑) | **0.952** | **0.952** | 0.952 |
| Kick (AUC ↑) | 0.767 | 0.766 | **0.770** |
| Breast Cancer (AUC ↑) | $0.992 \pm 0.008$ | $0.996 \pm 0.006$ | $\mathbf{0.997 \pm 0.001}$ |
| **Regression** | | | |
| Boston Housing (RMSE ↓) | $3.22 \pm 0.25$ | $3.18 \pm 0.35$ | $\mathbf{2.92 \pm 0.15}$ |
| Yacht (RMSE ↓) | $1.15 \pm 0.11$ | $\mathbf{0.50 \pm 0.06}$ | $1.27 \pm 0.15$ |
| Concrete (RMSE ↓) | $\mathbf{4.79 \pm 0.12}$ | $5.37 \pm 0.20$ | $5.21 \pm 0.20$ |
| Protein (RMSE ↓) | **3.29** | 3.59 | 3.41 |

## B.5 Ablation Study 2: NPT without ABA and NPT without Feature Masking

We next present an additional ablation study targeting two core components of NPTs across all datasets: the Attention Between Attributes (ABA) layer and the stochastic feature masking.

**ABA Layer.** First, we perform an ablation to test if ABA layers are beneficial in practice. For this, we simply leave out the ABA layers, such that the MLP at the end of the ABD layers (see "rFF" in Eq. (5)) is now the only way for the model to independently transform the features of input datapoints.

Our results, given in Table 5, show that, generally, ABA is a useful component of the NPT architecture. Leaving out ABA increases performance only for 3/10 datasets. Interestingly, all three of these datasets are regression tasks, which may warrant further investigation. We observe the largest difference for the Poker Hands dataset, which requires complex reasoning between input features: in the same number of training steps, the ablation only achieves 57.4% accuracy compared to 99.3% for full NPT. These results support our hypothesis that ABA is useful when the dataset requires complex transformations of the features. Our most general recommendation would be to default to using NPTs with ABA layers, as they boost performance on the majority of datasets we examine. However, if practitioners can spend the extra compute, exploring NPTs without ABA can be worthwhile.

**Stochastic Feature Masking.** We perform an ablation to test if the stochastic feature masking objective (cf. §2.6) is beneficial in practice. For this, we simply disable all stochastic masking of input features by setting $p_{\text{features}} = 0$.

Our results, also in Table 5, show that for 9/10 datasets, enabling feature masking yields at least a small improvement in performance. Disabling feature masking is detrimental to the performance on the Poker Hands dataset, leading to a 30% drop in accuracy. Again, our general recommendation would be to use NPTs with feature masking by default, as it rarely seems to decrease performance and sometimes helps significantly, but to explore NPTs without feature masking if feasible.

## B.6 Computational Cost of Non-Parametric Transformers

We next compare the computational requirements of NPT against the various baselines. More specifically, we compare experiment runtimes and maximum memory usage on the Protein and Higgs datasets. We choose these datasets because they are representative of medium and large datasets in terms of computational requirements, with $45\,730$ and $11\,000\,000$ datapoints respectively. Note that, while we re-use hyperparameter configurations across datasets for NPTs, the baselines require a novel hyperparameter search to be performed for each dataset (cf. Appendices C and E). Below, we include the cost of hyperparameter optimization for the baselines.

Note that these numbers only provide a rough ordering of the compute and memory costs of the various methods. We did *not* optimize the baselines or NPT for memory usage, training time, or prediction speed. Additionally, while NPTs rely on GPU-accelerated PyTorch code, many of the baselines are CPU-only: therefore, the results depend on our particular CPU and GPU choices.

We also give the number of CPUs used in each experiment for each baseline. Here, we maximize the number of CPUs used in parallel execution in order to speed up training. This is mainly limited by the memory used per process: e.g., if we list # CPUs as 1, this does not mean that we used a machine with only 1 CPU, but rather that each process used a significant amount of the total available memory and hence we could not increase the number of CPUs used in parallel. Note that, additionally, for the CPU baselines, we made use of high-memory instances when this was necessary to avoid out-of-memory issues.

In summary, the numbers we give are a rough indication of the computational cost that a practitioner should expect to require in order to reproduce our results. It is likely that by tuning aspects of our setup, both for NPTs and the baselines, memory usage and/or runtimes could be improved.

We display the observed computational costs in Tables 6 and 7 for the Protein and Higgs datasets. As of now, NPTs do generally require longer training times than the non-neural baselines. For example, for the Protein dataset, the selected hyperparameter configuration of NPT trains in 11 hours, while all boosting methods finish their runs in less than 1 hour, including the hyperparameter tuning. The exception to this rule is given by some of the baselines, e.g., Random Forests, which do not scale well to large datasets such as Higgs. On Higgs, the NPT run takes 5d 22h compared to 13d 13h for Random Forests.

With NPTs, we want to store as much data as possible in addition to the network weights; recall that this is done to improve the quality of the minibatch approximation of the full dataset. Therefore, as expected, NPT is much more GPU-memory intensive during training than TabNet, the only other baseline with a GPU-based implementation, for which maximizing minibatch size is not desirable. In particular, the peak GPU memory usage on Higgs for NPTs is 19.18 GB and 1.18 GB for TabNet. However, we note that other methods are often also memory-intensive on larger datasets. For example, Random Forest with 1 process uses 189.18 GB peak CPU memory.

We next give a rough indication of prediction time behavior of NPT and the baselines. For the same reason as above, NPT is expected to have high memory usage at prediction time. In terms of prediction speed, we suspect that our ability to scale NPT to large batch sizes, e.g., 4096 on the Higgs dataset, might give us an advantage in comparison to those baselines that cannot be parallelized well and/or lack GPU support. We leave a detailed investigation of prediction time behavior to future work.

Finally, as discussed in §5, we note that by incorporating recent tools for sparse and efficient attention [5, 18, 19, 47, 84], future research could significantly improve the scalability of NPTs.

## B.7 Extended Results for Tabular Data Benchmarks

See Table 8 (Table 9) for test accuracies (negative log-likelihood scores) on the UCI classification datasets and additionally Table 10 for AUROC results on the binary classification datasets. For the regression datasets, see Table 11 for RMSE scores and Table 12 for MSE scores.

---

[8]Out-of-memory on the Higgs Boson dataset when attempting approximate 3-NN on an Azure D64 v3 instance with 256 GB RAM.

[9]TabNet had notably lower accuracy in our setup on the Poker Hand dataset (which has a fixed test set) than that the 99.2% reported in the original work [2]. We are in communication with the authors, attempting to improve these results. However, our results on Higgs Boson match the reported performance more closely (78.44% (theirs) vs 77.1% (ours)). Further, we note that our other baselines achieve significantly better performance on the same datasets than those reported in [2]; e.g., our MLP achieves 99.5% accuracy on Poker Hand dataset while they report 50.0%; our XGBoost achieves 97.1% on Forest Cover while they report 89.34%. However, we note that some of the datasets – such as Forest Cover – do not have fixed test sets. Therefore, we cannot exclude the possibility that the performance differences are due to differently chosen train-test splits.

[10]See above note on out-of-memory.

[11]See above note on out-of-memory.

Table 6: Protein dataset (45,730 datapoints): compute and memory requirements of hyperparameter tuning for baselines and training time of the selected hyperparameter configuration for NPTs. We report the number of CPUs used in execution, execution time, and peak memory usage, where the relevant bottleneck is main memory usage for CPU-based methods and GPU memory usage for GPU-based methods (i.e., TabNet and NPT).

| *Metric* | # CPUs | Execution Time | Peak Main Memory (GB) | Peak GPU Memory (GB) |
|---|---|---|---|---|
| Random Forest | 8 | 13h 33m 58s | 7.82 | – |
| Gradient Boosting | 1 | 47m 51s | 11.17 | – |
| XGBoost | 8 | 10m 31s | 2.94 | – |
| CatBoost | 1 | 8m 33s | 11.27 | – |
| LightGBM | 8 | 21s | 1.65 | – |
| MLP | 64 | 42m 14s | 8.96 | – |
| k-NN | 8 | 1m 8s | 40.47 | – |
| TabNet | 1 | 1h 33m 35s | 16.00 | 3.72 |
| NPT | 4 | 11h 51m 25s | 4.42 | 6.17 |

Table 7: Higgs dataset (11,000,000 datapoints): compute and memory requirements of hyperparameter tuning for baselines and training time of the selected hyperparameter configuration for NPTs. We report the number of CPUs used in execution, execution time, and peak memory usage, where the relevant bottleneck is main memory usage for CPU-based methods and GPU memory usage for GPU-based methods (i.e., TabNet and NPT).

| *Metric* | # CPUs | Execution Time | Peak Main Memory (GB) | Peak GPU Memory (GB) |
|---|---|---|---|---|
| Random Forest | 1 | 13d 13h 5m 6s | 189.18 | – |
| Gradient Boosting | 1 | 3d 19h 45m 56s | 26.65 | – |
| XGBoost | 8 | 23h 26m 17s | 108.54 | – |
| CatBoost | 8 | 2h 6m 35s | 78.34 | – |
| LightGBM | 8 | 55m 57s | 35.13 | – |
| MLP | 6 | 12h 54m 7s | 34.41 | – |
| k-NN | 1 | 4d 22h 12m 20s | 16.26 | – |
| TabNet | 1 | 2d 5h 2m 43s | 16.00 | 1.18 |
| NPT | 4 | 5d 22h 12m 7s | 37.79 | 19.18 |

Table 8: UCI classification datasets: test accuracy. Standard error reported for datasets with multiple cross-validation splits.

| *Test Accuracy* ↑ | Higgs Boson | Poker Hand | Forest Cover | Income | Kick | Breast Cancer |
|---|---|---|---|---|---|---|
| Random Forest | 76.2 | 71.5 | 94.8 | 95.4 | 90.1 | $94.20 \pm 0.70$ |
| Gradient Boosting | 76.5 | 94.1 | 96.7 | 95.8 | 90.2 | $94.03 \pm 0.90$ |
| XGBoost | 77.0 | 95.9 | **97.1** | 95.6 | **90.3** | $94.91 \pm 0.68$ |
| CatBoost | 76.6 | 99.2 | 95.7 | **95.8** | 90.1 | $\mathbf{95.61 \pm 0.75}$ |
| LightGBM | 75.9 | 92.8 | 85.0 | **95.8** | **90.3** | $95.26 \pm 0.82$ |
| MLP | 78.3 | **99.5** | 95.2 | 95.4 | 90.0 | $94.73 \pm 0.89$ |
| k-NN[8] | — | 50.4 | 90.7 | 94.8 | 87.7 | $95.26 \pm 0.79$ |
| TabNet[9] | 77.1 | 53.3 | 94.2 | 95.5 | 89.5 | $94.91 \pm 0.76$ |
| **NPT** | **80.7** | 99.3 | 96.7 | 95.6 | 90.0 | $94.73 \pm 0.69$ |

Table 9: UCI classification datasets: negative log-likelihood (NLL). Standard error reported for datasets with multiple cross-validation splits.

| Test NLL ↓ | Higgs Boson | Poker Hand | Forest Cover | Income | Kick | Breast Cancer |
|---|---|---|---|---|---|---|
| Random Forest | 0.489 | 0.843 | 0.191 | 0.126 | 0.305 | $0.142 \pm 0.012$ |
| Gradient Boosting | 0.477 | 0.379 | 0.109 | 0.111 | 0.296 | $0.185 \pm 0.024$ |
| XGBoost | 0.471 | 0.178 | **0.080** | 0.147 | **0.293** | $0.143 \pm 0.025$ |
| CatBoost | 0.476 | 0.065 | 0.120 | **0.109** | 0.296 | $\mathbf{0.124 \pm 0.024}$ |
| LightGBM | 0.486 | 0.420 | 0.361 | **0.109** | 0.294 | $0.163 \pm 0.034$ |
| MLP | 0.452 | **0.028** | 0.131 | 0.118 | 0.333 | $0.545 \pm 0.254$ |
| k-NN[10] | — | 0.975 | 0.274 | 0.139 | 0.333 | $0.466 \pm 0.167$ |
| TabNet | 0.469 | 0.973 | 0.151 | 0.119 | 0.314 | $0.233 \pm 0.036$ |
| **NPT** | **0.412** | 0.119 | 0.087 | 0.115 | 0.299 | $0.137 \pm 0.026$ |

Table 10: UCI classification datasets: test area under the receiver operating characteristic curve (AUROC) on binary classification tasks. Standard error reported for datasets with multiple cross-validation splits.

| Test AUROC ↑ | Higgs Boson | Income | Kick | Breast Cancer |
|---|---|---|---|---|
| Random Forest | 0.847 | 0.947 | 0.759 | $0.989 \pm 0.003$ |
| Gradient Boosting | 0.850 | 0.955 | 0.769 | $0.987 \pm 0.004$ |
| XGBoost | 0.854 | 0.946 | 0.775 | $0.989 \pm 0.003$ |
| CatBoost | 0.851 | **0.956** | 0.773 | $0.992 \pm 0.003$ |
| LightGBM | 0.843 | **0.956** | **0.776** | $0.992 \pm 0.003$ |
| MLP | 0.867 | 0.949 | 0.739 | $0.982 \pm 0.007$ |
| k-NN[11] | — | 0.932 | 0.747 | $0.980 \pm 0.005$ |
| TabNet | 0.857 | 0.948 | 0.745 | $0.978 \pm 0.005$ |
| **NPT** | **0.892** | 0.952 | 0.770 | $\mathbf{0.997 \pm 0.001}$ |

Table 11: UCI regression datasets: test root mean-squared error (RMSE). Standard error reported for datasets with multiple cross-validation splits.

| Test RMSE ↓ | Protein | Concrete | Boston Housing | Yacht |
|---|---|---|---|---|
| Random Forest | 3.57 | $5.48 \pm 0.18$ | $3.78 \pm 0.33$ | $0.91 \pm 0.13$ |
| Gradient Boosting | 3.61 | $4.70 \pm 0.18$ | $3.44 \pm 0.22$ | $\mathbf{0.85 \pm 0.12}$ |
| XGBoost | 3.60 | $4.68 \pm 0.15$ | $3.39 \pm 0.29$ | $0.88 \pm 0.13$ |
| CatBoost | 3.51 | $\mathbf{4.28 \pm 0.16}$ | $3.44 \pm 0.34$ | $1.05 \pm 0.16$ |
| LightGBM | 3.65 | $4.64 \pm 0.18$ | $3.86 \pm 0.27$ | $13.60 \pm 0.73$ |
| MLP | 3.62 | $5.53 \pm 0.20$ | $3.32 \pm 0.39$ | $0.91 \pm 0.13$ |
| k-NN | 3.77 | $8.51 \pm 0.30$ | $4.27 \pm 0.37$ | $12.02 \pm 0.65$ |
| TabNet | 3.59 | $5.85 \pm 0.15$ | $3.88 \pm 0.34$ | $3.41 \pm 1.12$ |
| **NPT** | **3.41** | $5.21 \pm 0.20$ | $\mathbf{2.92 \pm 0.15}$ | $1.27 \pm 0.15$ |

Table 12: UCI regression datasets: test mean-squared error (MSE). Standard deviation reported for datasets with multiple cross-validation splits.

| Test MSE ($\pm$ Std Dev) $\downarrow$ | Protein | Concrete | Boston | Yacht |
|---|---|---|---|---|
| Random Forest | 12.8 | $30.4 \pm 6.4$ | $15.4 \pm 9.5$ | $0.986 \pm 0.818$ |
| Gradient Boosting | 13.0 | $22.4 \pm 5.2$ | $12.3 \pm 4.9$ | $\mathbf{0.867 \pm 0.779}$ |
| XGBoost | 13.0 | $22.1 \pm 4.2$ | $12.3 \pm 7.6$ | $0.939 \pm 0.881$ |
| CatBoost | 12.3 | $\mathbf{18.6 \pm 4.3}$ | $13.0 \pm 9.8$ | $1.36 \pm 1.12$ |
| LightGBM | 13.3 | $21.9 \pm 5.3$ | $15.6 \pm 7.6$ | $190.0 \pm 65.1$ |
| MLP | 13.1 | $31.0 \pm 6.9$ | $12.6 \pm 11.0$ | $0.994 \pm 0.937$ |
| k-NN | 14.2 | $73.3 \pm 16.0$ | $19.6 \pm 11.0$ | $149.0 \pm 52.6$ |
| TabNet | 12.9 | $34.4 \pm 5.8$ | $16.2 \pm 11.0$ | $24.1 \pm 54.3$ |
| **NPT** | **11.6** | $27.6 \pm 7.6$ | $\mathbf{8.77 \pm 2.60}$ | $1.80 \pm 1.49$ |

## B.8 Image Classification Results

We explore two different setups for applying NPTs to high-dimensional image data: (1) using a CNN encoder based on the ResNet-18 architecture, followed by ABD layers, and (2) using a linear patching encoder that is then followed by ABD and ABA layers. We present results using (1) for CIFAR-10 and (2) for MNIST in the main paper, and additionally provide results using (2) on CIFAR-10 below.

Note that the aim of our image classification experiments is not to match the performance of a pretrained Transformer image classifier. Rather, we hope to demonstrate that NPTs can readily learn interactions between datapoints on a wide variety of data modalities and tasks, including image classification, while achieving reasonable performance.

**(1) CNN Encoder.** In this setup, we replace our linear encoder with a CNN, which is then folowed by several rounds of Attention Between Datapoints (ABD) on the CNN encodings. We apply this setup to CIFAR-10.

In detail, we use a ResNet-18 encoder followed by 4 blocks of ABD (as we have in a default 8 layer NPT, cf. Appendix C.1.1) with 8 heads. Because we do not use Attention Between Attributes (ABA) the output of the encoder corresponds to the dimensions $h = d \cdot e = 128$. We train in a supervised manner (without test inputs available at training time) with a training batch size of 128 and evaluation batch size of 480, for a fixed 100 epochs.

As reported in the main text, we achieve a test accuracy of 93.7% on CIFAR-10 with this architecture. We find that the data corruption test (cf. Section 4.3) decreases accuracy by 1.2%, which suggests that NPT meaningfully relies on other datapoints for prediction on CIFAR-10. The ResNet-18 alone achieves a test accuracy of 93.9%.

We further note that with a ResNet-18 encoder pretrained on ImageNet, our ResNet + NPT architecture achieves a test accuracy of 94.7% on CIFAR-10, and loses 0.7% in the data corruption experiment, whereas the pretrained ResNet-18 alone achieves a lower 94.2% accuracy. We believe that an exploration of how pretraining might affect the performance of NPT and the extent to which predictions rely on other datapoints is interesting future work.

**(2) Linear Patching Encoder.** We additionally consider an image classification setup using a linear patching encoder, which we apply to both MNIST and CIFAR-10.

In detail, we append the mask dimension as an extra channel and apply image patching with linear embeddings as in [25]. Further following [25], we use a learned position embedding for each patch and the class token. We use $7 \times 7 = 49$ patches on MNIST and $8 \times 8 = 64$ patches on CIFAR-10. On both datasets, for this linear patching setup, we begin with the `NPT-Base` architecture described in C.1.1. On MNIST, we use batch size 512, train for 500,000 steps, use hidden dimensions $e = 16$, $p_{target} = 0.15$, and use $7 \times 7 = 49$ patches. On CIFAR-10, we use batch size 512, train for 1,000,000 steps, use random crops and horizontal flips for data augmentation, use $8 \times 8 = 64$ patches of each image, and do not use target masking due to constraints on compute time.

With this setup, NPT achieves 98.3% accuracy on MNIST and 68.2% accuracy on CIFAR-10. We additionally find in the data corruption experiment (detailed in Section 4.3) that after destroying information from other datapoints, the change in accuracy is -0.4% on MNIST and -5.1% on CIFAR-10, demonstrating that NPTs learn to make use of interactions between images.

However, we did not find that this sufficiently demonstrated that NPTs make use of datapoint interactions in achieving *reasonable* performance on CIFAR-10, and hence conducted the experiment on CIFAR-10 using the CNN encoder setup above.

We expect that the relatively low performance in the linear patching setup on CIFAR-10 was due to a number of differences between our setup and other works, which report state-of-the-art results on image classification using Transformers and linear patching. Most importantly, previous works [25, 46] either consider only, or pretrain on, large or huge datasets; for example, ImageNet [23, 46, 86], ImageNet-21k [75], or JFT-300M, with over 375 million labeled datapoints [25, 83]. We perform no pretraining, and therefore a direct comparison of these results to this line of work is inappropriate. Additionally, previous works use significantly more patches (e.g., 256 in [25]) and use higher resolutions, including during fine-tuning by upscaling from $32 \times 32$ to $224 \times 224$ resolution [25, 46, 54, 85].

# C Additional Details on the NPT Architecture

## C.1 NPT Training and Hyperparameters

### C.1.1 NPT-Base Architecture

Below, we outline the `NPT-Base` model configuration. The final configurations used for each dataset are essentially the same as `NPT-Base`, with minor alterations in parameters such as hidden dimension size, learning rate warmup, batch size, and number of training steps. Given our limited memory and compute time budget, these changes directly result from differences in number of datapoints/attributes between the datasets. We divide the `NPT-Base` configuration into architectural details and optimization details.

`NPT-Base` **Architecture**

- 8 layers, alternating Attention Between Datapoints and Attention Between Attributes.
- 8 heads.
- Row-wise feed-forward (rFF) networks with one hidden layer, 4x expansion factor, and GeLU activation (standard in Transformer literature [66, 90]).
- Attention weight and hidden layer dropout with $p = 0.1$ (cf. Appendix C.2.1).
- Per-attribute hidden dimension $e = 64$.

`NPT-Base` **Optimization**

- LAMB [98] optimizer with $\beta = (0.9, 0.999)$ and $\epsilon = 1e-6$, and a Lookahead [99] wrapper with slow update rate $\alpha = 0.5$ and $k = 6$ steps between updates.
- Stochastic feature masking probability $p_{\text{feature}} = 0.15$.
- Anneal the tradeoff $\lambda$ between feature and target loss with a cosine schedule, starting at 1 (all feature loss) to 0 (all target loss) over the course of training.
- Flat-then-anneal learning rate schedule: flat at the base learning rate for 70% of steps, and then anneals following a cosine schedule to 0 by the end of training.
- Base learning rate 1e-3.
- Gradient clipping at 1.

On all datasets with minibatching, we approximately maintain relative train, validation, and test datapoint proportions in each batch. We train NPT in semi-supervised mode (cf. Appendix C.4.2) but have found that this does not consistently improve performance compared to conventional training because the amount of unlabeled test data is usually comparatively small.

### C.1.2 NPT Training on Small Data

Here we describe the hyperparameter sweep details for small datasets – Breast Cancer, Boston, Concrete, and Yacht.

**Base Hyperparameter Configurations.** Across these small datasets, we make a few minor adjustments to the `NPT-Base` architecture and optimization to obtain the `NPT-Small` configuration: we increase the default number of hidden dimensions to $e = 128$, fix the flat-then-anneal schedule to be flat for 50% instead of 70% of steps, and train with the entire dataset as input, i.e., no minibatching. We set stochastic target masking probability to $p_{\text{target}} = 1$ by default, i.e., deterministically mask out train labels as would be done in a normal supervised setting, and then introduce modifications in our sweep.

Note that the vast majority of hyperparameters such as the number of layers and heads, optimizer, $p_{\text{feature}}$, tradeoff annealing schedule, learning rate schedule, and gradient clipping are exactly the same between `NPT-Base` and `NPT-Small`.

We would like to keep the base configuration for each of the small datasets exactly the same. However, we need to slightly vary the learning rate and number of epochs per dataset to optimize loss convergence across datasets. We use a base learning rate 5e-4 on Breast Cancer and 1e-3 on the other small datasets. We train for 2000 epochs on Breast Cancer and Boston, and $10\,000$ epochs on Yacht and Concrete. On Breast Cancer, we additionally drop $e = 32$ due to memory constraints (it has more attributes than other small datasets).

**Small Data Sweep.** Based on these configurations, we sweep over the following 8 configurations of the model on each dataset.[12]

- Vanilla `NPT-Small` model for given dataset.
- Increase number of layers $8 \rightarrow 16$.
- Increase number of heads $8 \rightarrow 16$.
- Increase number of layers $8 \rightarrow 16$, and number of heads $8 \rightarrow 16$.
- Stochastic target masking with probability $p_{\text{target}} = 0.1$.
- Stochastic target masking with probability $p_{\text{target}} = 0.5$.
- Increase stochastic feature masking probability from 0.15 to $p_{\text{feature}} = 0.2$.
- Use a cosine cyclic learning rate scheduler with two cycles, initial learning rate 1e-7, final learning rate 1e-7, and max learning rate given by the base model learning rate.

For the stochastic target masking variants, we proportionally increase the number of epochs (e.g., with $p_{\text{target}} = 0.5$, half as many targets are observed in a given epoch, so we double the total number of epochs).

**Small Data Variant Rank Orders.** We report the rank order ($\pm$ standard error) of these variants in Table 13. A notable trend is that the *target masking configurations perform particularly well*. One of the two configurations with target masking is the top performer on each of the four datasets. This could be attributed to some combination of the representational advantage of label masking (cf. Section 2.6), an additional regularization effect akin to dropout, or stabler convergence over a greater number of epochs.

Other configurations did not display similarly obvious trends in performance. This is in concordance with the ablation study (Appendix B.4) and supports the claim that NPT is robust to changes in hyperparameters.

### C.1.3 NPT Training on Medium and Large Data

For the medium and large datasets, we again adopt the `NPT-Base` architecture and optimization hyperparameters, and make minor manual changes on a per-dataset basis to account for differences in number of datapoints and attributes across the datasets. No more than 3 manual iterations are performed to find these adaptations. We generally attempt to maximize batch size given a fixed memory budget. Given the rank order results on small data (cf. Table 13) we use target masking on the medium and large datasets whenever computationally feasible.[13] These per-dataset alterations are reported below.

---

[12]Note that we do not search a $2^8$ grid over these modifications. We only try out these 8 distinct models.

[13]Training is slower as only a $p_{\text{target}}$ proportion of training labels are used for backpropagation in each epoch. Therefore, target masking may increase training time beyond our budget.

Table 13: Average rank order of variants of NPT-Small ($\pm$ standard error) across 10 cross-validation splits on each small dataset. We determine rank using negative log-likelihood and sort methods by ascending rank for each metric.

| *Dataset* | Boston |
| --- | --- |
| $p_{\text{target}} = 0.5$ | $2.50 \pm 0.73$ |
| $p_{\text{target}} = 0.1$ | $2.50 \pm 0.83$ |
| $8 \rightarrow 16$ Layers, $8 \rightarrow 16$ Heads | $2.60 \pm 0.65$ |
| Cosine Cyclic LR Schedule | $3.10 \pm 0.75$ |
| Base NPT-Small | $3.70 \pm 0.84$ |
| $8 \rightarrow 16$ Layers | $4.30 \pm 0.67$ |
| $8 \rightarrow 16$ Heads | $4.40 \pm 0.60$ |
| $p_{\text{feature}} = 0.2$ | $4.90 \pm 0.46$ |

| *Dataset* | Breast Cancer |
| --- | --- |
| $p_{\text{target}} = 0.1$ | $2.60 \pm 0.92$ |
| Base NPT-Small | $2.70 \pm 0.65$ |
| $8 \rightarrow 16$ Heads | $3.00 \pm 0.49$ |
| $p_{\text{feature}} = 0.2$ | $3.20 \pm 0.68$ |
| $p_{\text{target}} = 0.5$ | $3.50 \pm 0.56$ |
| Cosine Cyclic LR Schedule | $4.10 \pm 0.89$ |
| $8 \rightarrow 16$ Layers, $8 \rightarrow 16$ Heads | $4.40 \pm 0.70$ |
| $8 \rightarrow 16$ Layers | $4.50 \pm 0.81$ |

| *Dataset* | Concrete |
| --- | --- |
| $p_{\text{target}} = 0.5$ | $2.30 \pm 0.76$ |
| Cosine Cyclic LR Schedule | $2.50 \pm 0.69$ |
| $8 \rightarrow 16$ Heads | $2.60 \pm 0.62$ |
| Base NPT-Small | $2.70 \pm 0.52$ |
| $p_{\text{target}} = 0.1$ | $3.10 \pm 0.64$ |
| $8 \rightarrow 16$ Layers | $3.90 \pm 0.80$ |
| $8 \rightarrow 16$ Layers, $8 \rightarrow 16$ Heads | $5.10 \pm 0.66$ |
| $p_{\text{feature}} = 0.2$ | $5.80 \pm 0.39$ |

| *Dataset* | Yacht |
| --- | --- |
| $p_{\text{target}} = 0.1$ | $1.20 \pm 0.53$ |
| $p_{\text{target}} = 0.5$ | $2.70 \pm 0.52$ |
| $8 \rightarrow 16$ Heads | $2.80 \pm 0.66$ |
| Cosine Cyclic LR Schedule | $3.10 \pm 0.69$ |
| Base NPT-Small | $3.60 \pm 0.54$ |
| $8 \rightarrow 16$ Layers | $4.10 \pm 0.74$ |
| $p_{\text{feature}} = 0.2$ | $5.20 \pm 0.47$ |
| $8 \rightarrow 16$ Layers, $8 \rightarrow 16$ Heads | $5.30 \pm 0.83$ |

**UCI Datasets.** We report results for Protein using the Base NPT configuration in the ablation study (cf. Table 4). On Kick, we use batch size 4096, train for $250\,000$ steps, and use $p_{\text{target}} = 0.5$. On Income, we use batch size 2048, train for $2\,000\,000$ steps, use no feature masking (and correspondingly fix the tradeoff parameter $\lambda = 0$), and use $p_{\text{target}} = 0.15$. On Poker Hand, we use batch size 4096, train for $200\,000$ steps, use $p_{\text{target}} = 0.5$, and stratify by class (i.e., compose training datapoints in each minibatch proportionally to the empirical label distribution of the training set to account for significant class imbalance). On Forest Cover, we use batch size 1800, train for $800\,000$ steps, use a polynomial decay learning rate scheduler with warmup over the first $1\%$ of steps, use base learning rate 0.005, $p_{\text{target}} = 0.5$, and class balancing as above. The changes to learning rate scheduling were made to speed up training and hence save compute resources. On Higgs, we use batch size 4096, train for $500\,000$ steps, and do not use target masking due to constraints on compute time.

**Image Data (CIFAR-10 and MNIST).** See Appendix B.8 for details on the image data architecture and setup.

Again, we stress that the vast majority of hyperparameters used on all datasets (small, medium, and large benchmarks from UCI as well as the image benchmarks) are identical; configurations follow NPT-Base (cf. Appendix C.1.1) very closely and changes usually affect NPT optimization rather than architecture.

## C.2   Further Details on ABD and ABA Layers

### C.2.1   Dropout

In practice, we apply elementwise dropout on the attention scores $\exp(\boldsymbol{Q}\boldsymbol{K}^{\top}/\sqrt{h})$, as well as on the input/output embeddings and the output of the MHSelfAtt$(\cdot)$ function (often referred to as attention and hidden dropout).

## C.3 Input and Output Embdedings

### C.3.1 Input Embedding

At a high-level, we embed inputs by encoding categorical attributes as one-hot vectors and standardizing continuous attributes, followed by a learned linear embedding for each attribute to obtain $\text{InputEmbed}(\boldsymbol{X}) = \boldsymbol{H}^{(0)} \in \mathbb{R}^{n \times d \times e}$.

More specifically, we perform the following sequence of steps: Attributes $\boldsymbol{X}_{:,j}, j \in \{1, \ldots, d\}$ of the input matrix can be either continuous or categorical. We first apply a function $\text{Encode}(\cdot)$ to each attribute $\boldsymbol{X}_{:,j}$. This "encodes" categorical attributes with a one-hot representation and standardizes continuous attributes to zero mean and unit standard deviation. Each encoded attribute $j$ has (potentially unique) dimensions $n \times e_j$. Then, we concatenate this encoded attribute with its respective column of the masking matrix $\boldsymbol{M}_{:,j}$ along the second dimension to produce a column encoding of dimensions $n \times (e_j + 1)$. We learn separate embedding weights for each attribute $\boldsymbol{W}_j^{\text{in}} \in \mathbb{R}^{(e_j+1) \times e}$ that embed all attributes to a common hidden dimension $e$. Altogether, we can state the embedding of a single attribute column $\boldsymbol{X}_{:,j}$ as

$$\boldsymbol{H}_{:,j}^{(0)} = \underset{\text{axis}=e}{\text{concat}}(\text{Encode}(\boldsymbol{X}_{:,j}), \boldsymbol{M}_{:,j})\boldsymbol{W}_j^{\text{in}} + \boldsymbol{H}_{:,j}^{\text{Index}} + \boldsymbol{H}_{:,j}^{\text{Type}}, \tag{22}$$

where $\boldsymbol{H}_{:,j}^{\text{Index}} \in \mathbb{R}^{n \times e}$ is a learnt embedding for the index and $\boldsymbol{H}_{:,j}^{\text{Type}} \in \mathbb{R}^{n \times e}$ for the type (either continuous or categorical) of attribute $j$.

Finally, we write the full NPT input embedding layer as

$$\text{InputEmbed}(\boldsymbol{X}) = \underset{\text{axis}=d}{\text{stack}}(\boldsymbol{H}_{:,1}^{(0)}, \ldots, \boldsymbol{H}_{:,d}^{(0)}) = \boldsymbol{H}^{(0)} \in \mathbb{R}^{n \times d \times e}. \tag{23}$$

The stack operation constructs $\boldsymbol{H}^{(0)} \in \mathbb{R}^{n \times d \times e}$ from $d$ attribute embeddings $\boldsymbol{H}_{:,j}^{(0)} \in \mathbb{R}^{n \times e}, j \in \{1, \ldots d\}$.

### C.3.2 Output Embedding

For an NPT with $L$ layers, we obtain an output prediction by applying a learnt linear output embedding (that closely mirrors the process of the input embedding) to the output of the last attention layer $\boldsymbol{H}^{(L)}$. We write the output embedding layer as

$$\text{OutputEmbed}(\boldsymbol{H}^{(L)}) = [\boldsymbol{Z}_{:,1}, \ldots, \boldsymbol{Z}_{:,d}] = \boldsymbol{Z}, \tag{24}$$

$$\text{where } \boldsymbol{Z}_{:,j} = \boldsymbol{H}_{:,j,:}^{(L)}\boldsymbol{W}_j^{\text{out}}. \tag{25}$$

Our prediction $\boldsymbol{Z}$ is a list of $d$ attribute predictions $\boldsymbol{Z}_j \in \mathbb{R}^{n \times e_j}$. We learn output embedding weights $\boldsymbol{W}_j^{\text{out}} \in \mathbb{R}^{e \times e_j}$ which are applied on attribute slices $\boldsymbol{H}_{:,j,:}^{(L)} \in \mathbb{R}^{n \times e}$ of the output of the $L$th layer $\boldsymbol{H}^{(L)} \in \mathbb{R}^{n \times d \times e}$. Note that the second dimension of each attribute prediction $\boldsymbol{Z}_j$ is determined by the encoding size (i.e., $e_j = 1$ for continuous attributes, $e_j$ is the number of categories for a categorical attribute) as in the input embedding. Note also that we do not predict a mask value (i.e., we do not predict to dimensions $n \times (e_j + 1)$ for each attribute). To obtain the final prediction matrix $\hat{\boldsymbol{X}} \in \mathbb{R}^{n \times d}$ we take the $\arg\max$ over the categorical predictions.

## C.4 NPT Masking

### C.4.1 Handling Missing Values

Real-world data – particularly tabular data – often contains *missing entries*. Many popular models for supervised prediction on tabular data cannot accommodate missing values as input. Instead they require that missing features are *imputed*, i.e., an additional model predicts a surrogate value for what the missing values could have been, such that the supervised model then receives a "clean" dataset as input which no longer overtly contains missing values.

For example, all scikit-learn [69] predictors, including Gradient Boosting and Random Forests, require an explicit imputation step before training. Often, extremely simple imputation methods are

used in practice. For example, TabNet [2] drops datapoints with >10% missing entries and otherwise applies univariate mean imputation as part of a Google AI Platform pipeline [70]; and CatBoost [71] treats a missing continuous entry as the minimum or maximum of that feature (univariate min/max imputation), or raises an error. While more complex imputation methods could in theory be applied as pre-processing [43, 50, 81, 82, 88], there will always remain a separation between the imputation step and the prediction model. Additionally, more complex imputation methods often require training and hyperparameter selection, such that the combined imputation and prediction process becomes cumbersome. Both for practical as well as performance reasons, it is desirable to have a single model that can *directly* handle missing data, learn complex internal imputation operations from the data, and at the same time learn the desired predictive function from features to target.

This is exactly what NPTs achieve. They are able to accommodate inputs with missing values gracefully without requiring any imputation pre-processing steps, therefore modeling data with missing values end-to-end. We can explicitly indicate that a value $X_{i,j}$ is missing by simply setting the mask token $M_{i,j} = 1$. Already in standard NPTs, the stochastic feature masking during training teaches NPTs to predict values for which $M_{i,j} = 1$ while ignoring the value of their entry $X_{i,j}$ at input. Further, no choice of fixed imputation algorithm has to be made with NPTs. Instead, NPTs learn directly from the data how to make predictions given missing values. Attention between datapoints might be particularly useful for learning a general mechanism of how to impute missing values by attending to other datapoints. We therefore suspect that NPTs could be a strong contender for predicting on data with missing values. Further, unlike common imputation pre-processing, NPTs do not discard the information of *which* attributes were missing. Future work could also explore the ability of NPT to model arbitrary correlations underlying the pattern of which data is missing, i.e., datasets where values are not missing at random.

### C.4.2 Masking Encompasses Many Common Machine Learning Settings

The flexible masking mechanism of NPTs can be used to accommodate a variety of common machine learning settings.

**Multi-Target Prediction.** In *multi*-target classification or regression, more than one column of the dataset contains targets. Standard supervised models often do not support multi-output settings and must resort to training multiple models, one for each target. NPTs can accommodate multi-target prediction trivially, since they learn to make predictions at any masked input entry. For prediction in a multi-target setting, we simply apply target masking on all columns with targets.

**Self-Supervision.** In self-supervised learning, we are often interested in learning a generative model or useful encoding from unlabeled data. The reconstruction of corrupted input features as part of stochastic feature masking can already be seen as self-supervised learning. The stochastic masking mechanism allows NPTs to learn to predict masked out values anywhere in the input. In theory, NPTs should be able to learn a fully generative model of the dataset in this manner.

**Semi-Supervision.** In semi-supervised learning, we hope to use large quantities of unlabeled data to aid in learning a predictive function on a small set of labeled data. Often, this involves a two-step process, such as learning a powerful autoencoder from all data and then training a predictor using the learnt encoder and the small set of labeled data. NPTs can accommodate semi-supervised learning without changes to the architecture. Specifically, we can include large amounts of unlabeled data by simply appending those feature values to the labeled input dataset. We indicate that no labels are available for all unlabeled datapoints $i'$ by setting their mask token at the target column $X_{i',d} = 1$. NPTs can use attention between datapoints to make use of information from the features of the unlabeled datapoints.

**Imputation.** With imputation, we refer to scenarios where the main task is to predict missing values for arbitrary attributes and datapoints. Similar to self-supervision, NPTs already learn how to do this from the stochastic masking mechanism that is enabled by default. (Unlike for the self-supervision category, the imputation scenario assumes that there are actually some missing values that we would like to predict.)

### C.4.3 Stochastic Masking: Details

For stochastic masking, a specified proportion of training entries (we default to 15% following [24]) are selected for masking at the start of each epoch. Among those entries chosen, we mask out the

value with 90% probability and randomize it with 10% probability. "Masking out" means that the original value $X_{i,j}$ is overwritten with zeros and the mask token is set to 1. Randomization is done for categorical targets by sampling a new class uniformly at random. Continuous targets are sampled from a standard Normal $\mathcal{N}(0, 1)$.

This sampling scheme is applied for both stochastic feature masking and stochastic target masking, where we allow for different masking proportions between the two ($p_{\text{feature}}$ and $p_{\text{target}}$). During training, a loss is backpropagated on the masked entries.

### C.5  NPT Optimization

Each of the losses $\mathcal{L}^{\text{Features}}$ (feature loss) and $\mathcal{L}^{\text{Targets}}$ (target loss) is normalized by the number of entries on which it is evaluated.

As described in Appendix C.1.1: we anneal the $\lambda$ parameter in the NPT objective using a cosine schedule, i.e., starting with full weight on the feature loss term at epoch 0 and annealing to full weight on the target loss term by the end of training. We use LAMB [98] with Lookahead [99] for optimization, which we find to perform well with large minibatches. We use a flat-then-anneal learning rate schedule with cosine decay, notable as Transformer works [24, 90] often report that a linear learning rate warmup is necessary for training stability. Our placement of Layer Normalization before self-attention ("pre-LayerNorm" [3, 16]) may contribute to our not needing this.

## D  Related Work – Continued

### D.1  Tree-Based Baselines

Tree-based approaches in machine learning have been popular for over half a century [11, 62, 64]. Each node of a tree splits the data into smaller subsets, and predictions are made at each of the leaves. The splits are learned from a set of training data by minimizing some objective function. Many established methods combine predictions of multiple trees through bagging [9] and/or boosting [78]. Bagging uses an ensemble of trees, each learned by training on a random subsample of the data. This approach is most popularly used in Random Forests [10]. Boosting learns a sequence of trees, conditioning the learning of each additional model on the predictions of previous models, with the aim of reducing overall prediction error.

Popular examples of tree-based boosting models include AdaBoost [31], XGBoost [17], CatBoost [71], and LightGBM [48]. To date, boosting arguably comprises the most popular approach for tabular data prediction. These models often rely on careful tuning of a large variety of hyperparameters. However, training cost is often cheap compared to neural network architectures, and therefore, so is hyperparameter optimization. This balance is slightly offset for NPTs, which seem largely robust to hyperparameter tuning. Hence, the training of a single NPT is often competitive to a grid search over hyperparameters for a tree-based model.

## E  Classification and Regression Benchmark Details

### E.1  General Setup

For certain datasets we use a canonical fixed test set. Otherwise, we default to 10-fold cross validation with 0.7/0.2/0.1 splits on smaller datasets and a single 0.7/0.1/0.2 split on larger datasets, where the exact split indices are always consistent across baselines. The full details on all UCI benchmark datasets are given in Tables 14 and 15. Note the variety of the datasets across number of instances, number of features, composition (categorical or continuous) of features, and task (multi-class classification, binary classification, and regression).

### E.2  Hyperparameter Tuning

#### E.2.1  Overview

Table 16 lists the number of unique hyperparameter configurations swept over for each baseline and classification/regression dataset.

Table 14: UCI classification dataset statistics and experimental setup details.

| Dataset | Higgs Boson | Poker Hand | Forest Cover | Income | Kick | Breast Cancer |
|---|---|---|---|---|---|---|
| # Instances | 11,000,000 | 1,025,010 | 581,012 | 299,285 | 72,983 | 569 |
| # Features | 28 | 10 | 54 | 42 | 32 | 31 |
| # Categorical Features | 0 | 10 | 44 | 36 | 18 | 0 |
| # Continuous Features | 28 | 0 | 10 | 6 | 14 | 31 |
| # Classes | 2 | 10 | 7 | 2 | 2 | 2 |
| Train/Val/Test Split | 0.84/0.12/0.05 | 0.017/0.003/0.98 | 0.7/0.1/0.2 | 0.57/0.1/0.33 | 0.7/0.1/0.2 | 0.7/0.2/0.1 |
| Fixed Test Set | Yes | Yes | No | Yes | No | No (10-Fold CV) |
| Uses Minibatching | Yes | Yes | Yes | Yes | Yes | No |

Table 15: UCI regression dataset statistics and experimental setup details.

| Dataset | Protein | Concrete | Boston | Yacht |
|---|---|---|---|---|
| # Instances | 45,730 | 1030 | 506 | 308 |
| # Features | 9 | 9 | 13 | 6 |
| # Categorical Features | 0 | 0 | 2 | 5 |
| # Continuous Features | 9 | 9 | 11 | 1 |
| Train/Val/Test Split | 0.7/0.1/0.2 | 0.7/0.2/0.1 | 0.7/0.2/0.1 | 0.7/0.2/0.1 |
| Fixed Test Set | No | No (10-Fold CV) | No (10-Fold CV) | No (10-Fold CV) |
| Uses Minibatching | Yes | No | No | No |

Table 16: Number of unique hyperparameter configurations swept over for each model class and dataset. Here we shorten Boston Housing to BH, Breast Cancer to BC, Poker Hand to PH, Forest Cover to FC, and Higgs Boson to HB. Datasets are ordered by increasing number of datapoints ($n$) from left to right.
* TabNet on Protein, Kick, and Income is tuned by sweeping over all 6 configurations listed in the original paper [2] in addition to the default configuration. Note that these configs include one tuned on Income.
† TabNet on Poker Hand, Forest Cover, and Higgs Boson use precisely the configuration specified for those datasets in the original paper [2].
‡ For some of these, we manually optimized convergence of the validation loss by adjusting non-architectural parameters such as learning rate (schedule), batch size, or number of steps in at most 3 iterations. See C.1.3.

| Dataset | Yacht | BH | BC | Concrete | Protein | Kick | Income | PH | FC | HB |
|---|---|---|---|---|---|---|---|---|---|---|
| Random Forest | 24 | 24 | 24 | 24 | 24 | 24 | 24 | 24 | 24 | 24 |
| Gradient Boosting | 48 | 48 | 48 | 48 | 48 | 48 | 48 | 48 | 48 | 48 |
| XGBoost | 48 | 48 | 48 | 48 | 48 | 48 | 48 | 48 | 48 | 48 |
| CatBoost | 48 | 48 | 48 | 48 | 48 | 48 | 48 | 48 | 48 | 48 |
| LightGBM | 48 | 48 | 48 | 48 | 48 | 48 | 48 | 48 | 48 | 48 |
| MLP | 11,340 | 11,340 | 11,340 | 11,340 | 270 | 270 | 270 | 270 | 270 | 6 |
| k-NN | 480 | 480 | 480 | 480 | 40 | 40 | 40 | 40 | 40 | - |
| TabNet | 48 | 48 | 48 | 48 | 7* | 7* | 7* | 1† | 1† | 1† |
| NPT | 8 | 8 | 8 | 8 | 1‡ | 1‡ | 1‡ | 1‡ | 1‡ | 1‡ |

All details on the NPT hyperparameter setup are given in Appendix C.1. Note that for any given dataset, NPT is tuned over fewer configurations than the baselines: we fix a base model configuration with minimal data-dependent tuning of hyperparameters such as learning rate, scheduler, number of steps, and target masking percentage $p_{\text{feature}}$, and choose the largest batch size viable for our hardware. On small datasets, we then sweep over 8 variants, and on medium and large datasets (including image data) use only the fixed variant with minor modifications.

In the case of TabNet, the configurations used for Poker Hand, Forest Cover, and Higgs Boson are those reported by the original authors for these datasets [2]; for Income, we performed a sweep over configurations including one reported for that dataset in the original publication. All deep learning approaches (MLP, TabNet, and NPT) use early stopping on the validation target loss.

### E.2.2 Baseline Sweep Details

We report hyperparameter sweep details for baselines below. The associated tables for each baseline give the bounds of the search space for numerical hyperparameters and all values for categorical hyperparameters. We clarify specific hyperparameters and provide context where helpful.

**Random Forest (Tables 17, 18).** `criterion` refers to the split criterion. `max_features` is the number of features to consider when looking for the best split.

**Gradient Boosting, XGBoost, LightGBM, and CatBoost (Table 19).** See D.1 for background on tree-based baselines.

**MLP (Tables 20, 21, 22).** The invscaling `learning_rate` scheduler scales with $\alpha_t = \alpha_0/t^{0.5}$ where $t$ is the step, $\alpha_0$ the initial learning rate, and $\alpha_t$ the learning rate at step $t$. The adaptive `learning_rate` divides the current learning rate by 5 when two consecutive epochs fail to decrease training or validation log loss by a tolerance 1e-4. Due to compute constraints, we decreased the size of the search space as the dataset size increased by focusing on 3-layer networks, lower L2 penalties, and higher batch sizes.

**k-NN (Tables 23, 24, 25).** `weights` describes the weight function applied to the neighborhood, i.e., "distance" means that closer neighbors of a query point have greater influence than those further away. `algorithm` specifies the underlying k-NN algorithm, where KD Tree [7] and Ball Tree [60] are approximations of brute-force search. The "auto" setting determines an appropriate algorithm based on the input data [69]. `leaf_size` is a hyperparameter of KD Tree and Ball Tree. `p` is the power parameter for the distance metric, i.e., `p` = 1 yields Manhattan and `p` = 2 Euclidean distance. It was computationally infeasible for us to obtain reasonable results on the 11M instance Higgs Boson dataset. Even when attempting approximate 3-NN on an Azure D64 v3 instance with 256 GB RAM, we encountered an out-of-memory error.

Table 17: Random Forest classification hyperparameters.

| *Hyperparameter* | criterion | n_estimators | max_features |
|---|---|---|---|
| Setting | gini, entropy | [50, 1000] | auto, sqrt, log2 |

Table 18: Random Forest regression hyperparameters.

| *Hyperparameter* | criterion | n_estimators | max_features |
|---|---|---|---|
| Setting | mae, mse | [50, 1000] | auto, sqrt, log2 |

Table 19: Gradient Boosting, XGBoost, LightGBM, and CatBoost hyperparameters (for both regression and classification).

| *Hyperparameter* | learning_rate | max_depth | n_estimators |
|---|---|---|---|
| Setting | [1e-3, 0.3] | [3, 10] | [50, 1000] |

Table 20: MLP hyperparameters for small datasets (Boston Housing, Breast Cancer, Concrete, and Yacht).

| *Hyperparameter* | hidden_layer_sizes | | l2_penalty |
|---|---|---|---|
| Setting | [(25)-(500), (25,25)-(500,500), (25,25,25)-(500,500,500)] | | [0, 1] |
| *Hyperparameter* | batch_size | learning_rate | learning_rate_init |
| Setting | [32, 256] | constant, invscaling, adaptive | [1e-5, 1e-1] |

Table 21: MLP hyperparameters for medium and large datasets other than Higgs Boson (Protein, Kick, Income, Poker Hand, Forest Cover).

| *Hyperparameter* | hidden_layer_sizes | l2_penalty |
|---|---|---|
| Setting | [(25,25,25)-(500,500,500)] | [0, 1e-2] |
| *Hyperparameter* | batch_size | learning_rate | learning_rate_init |
| Setting | [128, 256] | constant, invscaling, adaptive | [1e-5, 1e-1] |

Table 22: MLP hyperparameters for the Higgs Boson dataset.

| *Hyperparameter* | hidden_layer_sizes | l2_penalty |
|---|---|---|
| Setting | (500,500,500) | 0 |
| *Hyperparameter* | batch_size | learning_rate | learning_rate_init |
| Setting | [512, 1024] | constant | [1e-4, 1e-2] |

Table 23: k-NN hyperparameters for small datasets (Boston Housing, Breast Cancer, Concrete, and Yacht).

| *Hyperparameter* | n_neighbors | weights | algorithm | leaf_size | p |
|---|---|---|---|---|---|
| Setting | [2, 100] | uniform, distance | ball_tree, kd_tree, brute | [10, 100] | 1, 2 |

Table 24: k-NN hyperparameters for medium-large datasets (Protein, Kick, Income, Poker Hand).

| *Hyperparameter* | n_neighbors | weights | algorithm | leaf_size | p |
|---|---|---|---|---|---|
| Setting | [2, 1000] | distance | auto | [10, 100] | 2 |

Table 25: k-NN hyperparameters for Forest Cover.

| *Hyperparameter* | n_neighbors | weights | algorithm | leaf_size | p |
|---|---|---|---|---|---|
| Setting | [2, 25] | distance | auto | [10, 100] | 2 |

# F    Societal Impacts of NPT

We have introduced Non-Parametric Transformers, a novel deep learning architecture that predicts by including learned interactions between points of the dataset. In this work, we take first steps towards exploring NPTs and their properties. We do not recommend that NPTs are carelessly applied in production settings, because we do not yet know enough about them. We now list common concerns in applying machine learning models, discuss how they may apply to NPTs, and how to potentially mitigate them.

Many countries of the world, such as the US, UK, and the countries of the EU, are implementing "Right to Explanation"-schemes that grant those affected by autonomous decisionmaking the right to an explanation of why and how decisions were made. In general, Transformer-based architectures such as NPT have been shown to be amenable to explanations, see e.g., [90]. One could argue that our experiments in §4.4 move in an explanatory direction. However, we have not sufficiently investigated the explanations of individual NPT decisions, and believe this to be exciting future work.

Machine learning models are increasingly used in autonomous decision making that affects human beings in some capacity, e.g., clinical diagnosis, autonomous driving, and detection of toxic comments online.[14] It is of great importance that those decisions are *fair*, i.e., that they do not discriminate against underrepresented groups in some manner. We have not yet investigated how NPTs respond to common techniques of calibrating machine learning models to fulfil some definition of fairness. We believe that their special predictive behavior from similar datapoints likely poses both challenges and opportunities in this domain. For example, instead of needing to retrain the model to elicit changes in prediction – which could be infeasible in a real-world deployment – NPT could be "prompted" with a different set of context datapoints to modify its predictive behavior towards a more socially desirable response.

In large architectures based on Transformers, the memorization of training data is a common concern. If the model memorizes training data, adversarial attacks can be used to extract training data from the model weights, see e.g., [14]. This can lead to violations of privacy if, for example, a publicly available model was trained on data that must remain private. This can also cause more subtle problems; for example, if training data "lives on" in the model but must be deleted at some point in time to comply with privacy regulations. As NPT directly relies on training data as input for prediction, NPT is not a "private" model per definition. However, we can imagine future work tackling this question; for example, by learning to predict from a set of anonymous representative points instead of the training data directly.

At the model sizes presented in the paper, the environmental impact of training and using NPT is relatively small compared to some of the large architectures currently in fashion, see e.g., [12]. However, NPT could be scaled up to larger sizes at which point the energy used for training and prediction would become a serious concern. When considering tabular data, training a *single* NPT model is expensive compared to training a *single* one of our tree-based baselines such as XGBoost. However, we find that such baselines are often more sensitive to correctly tuned hyperparameters than NPT, such that the total compute including hyperparameter tuning of NPT and the baselines

---

[14]For example, see [28, 63, 87].

is actually often similar, particularly on larger datasets. Sparse approximations as referenced in Section 5 may further reduce the computational impact of NPT.

NPT is a new – and exciting – architecture. Therefore, in applications where explanations, fairness, or privacy are desired or legally required, we do not recommend that NPT be used at this stage.

# G   Code, Computational Resources, and License

**Code.**   We release code for NPTs at github.com/OATML/Non-Parametric-Transformers. The code-base relies on PyTorch [68] and NumPy [40], and we use Scikit-Learn [69] for many of the baseline experiments.

**Computational Resources.**   For the experiments we mainly rely on a shared internal cluster that has both NVIDIA Titan RTX GPUs (24 GB memory) as well as NVIDIA GeForce RTX 2080 Tis (12 GB memory). For tuning baselines, which are often compute-heavy workloads, we use Azure D-series compute-optimized VMs. For small datasets ($< 1000$ datapoints) such as Breast Cancer, training and evaluation of a single NPT model takes about 10 minutes. For larger datasets such as Protein ($< 100\,000$ datapoints), training and evaluation of NPT takes about 10 hours. For the largest datasets, e.g., Higgs Boson with 11 million datapoints, training and evaluation of NPT takes about 5 days. We did not optimize NPT for efficiency or training speed in this paper and suspect that convergence could be drastically improved with relatively little effort. The total amount of compute used for this paper is given by all NPT and baseline runs with repetitions for cross-validation, which amounts to more than 30 GPU days.

**License Agreements.**
*License agreement of the CIFAR-10 dataset*: CIFAR-10 is published under MIT license.

*License agreement of the MNIST dataset*: License: Yann LeCun and Corinna Cortes hold the copyright of MNIST dataset, which is a derivative work from original NIST datasets. MNIST dataset is made available under the terms of the Creative Commons Attribution-Share Alike 3.0 license.

*UCI Machine Learning Repository*:   Licenses for all datasets can be found at archive.ics.uci.edu/ml/.