# OpenReview forum: "Self-Attention Between Datapoints: Going Beyond Individual Input-Output Pairs in Deep Learning"
_NeurIPS.cc/2021/Conference — NeurIPS 2021 Poster_

### Official Review · Reviewer_bbzv · 2021-07-13

**Rating:** 7
**Confidence:** 4

**Summary:**

This paper proposes to use self-attention between data points, in order to model the relation between the samples in the dataset. The big claim of the paper is that by doing so, they are going beyond individual input-output pairs in deep learning. While true, this is the main paradigm of entire fields like person re-identification and metric learning, so this claim is massively overstated.

The paper presents good results in tabular datasets, reaching the best results in 4 out of 10 datasets, which is very good considering that different versions of boosting dominate the field.

**Limitations And Societal Impact:**

Limitations are discussed. The paper is a general framework, so the potential negative societal impact does not apply.

**Main Review:**

Strengths:

- The paper reaches very good results in tabular datasets. They reach the best results in 4 out of 10 datasets, better than any other method. When separated into the three main tasks: binary classification, multi-class classification, and regression, they reach the best results in binary classification, joined best results in multi-class classification, and joined second-best results in regression. Overall, it can be considered that the method works at least on par (in my opinion, slightly better) than the two main gradient boosting methods: XGBoost and CatBoost.

- Understanding the method is quite straightforward, the authors do a good job at explaining it.

- The authors do a good job at giving insights after they present their method.

Weaknesses:

1) Massively overstated claim. The first sentence of the paper 'We challenge a common assumption underlying most supervised deep learning: that a model makes a prediction depending only on its parameters and the features of a single input.' gives the impression that this is the first paper that tries to model some sort of relation between the data, instead of input-output relation which is commonly done in deep learning. However, this is a very misleading statement, contextual relations between the data in neural networks have been going on at least since RNNs were invented. More interestingly, there are entire fields in deep learning whose sole purpose is to model the relations between the data. Those fields fall under the umbrella of similarity learning, and are specialized in different directions like metric learning, person re-identification, face recognition etc.

In all those fields, the idea is that the embedding/results of a data point are not dependent only on itself, but also on the other data points. Typically that was done using some sort of contrastive/triplet loss, but nowadays that is done by using the relations between all the pairs in the dataset. That could be done as some sort of ranking, proxy, or contextual procedure. For example:

[A] Wang et al., Ranked List Loss for Deep Metric Learning, CVPR 2019 (Ranking)

[B] Movshovitz-Attias et al., No Fuss Distance Metric Learning using Proxies, ICCV 2017 (Proxy)

[C] Kim et al., Proxy Anchor Loss for Deep Metric Learning, CVPR 2020 (Proxy)

[D] Elezi et al., The Group Loss for Deep Metric Learning, ECCV 2020 (Contextual)

[E] Bao et al., Masked Graph Attention Network for Person Re-identification, CVPRW2019 (Contextual)

[F] Seidenschwarz et al., Learning Intra-Batch Connections for Deep Metric Learning, ICML 2021 (Contextual)

Take a look especially on papers [E, F]. They are doing almost the same thing as this paper, a rewriting of the GAT paper [G] to solve some other task. [F] specifically mentions that they use Transformers to model the relations between the samples in the dataset, which is the same thing as this paper.

[G] Velickovic et al., Graph Attention Networks, ICLR 2018.

Yet, this is not discussed at all in this paper (although [G] has been cited, but not properly discussed). In many ways, the paper is not doing anything more than these other papers, making the claim massively exaggerated.

2) Misleading statement -

line 62-63: We provide the model with the entire dataset – all datapoints – as input.
line 72: NPTs take as input the entire dataset $X \in R^{n×d}$.

My first impression was how is this scalable, and I was expecting a sophisticated way of somehow memorizing the dataset or using some hashing procedure that allows that. To my disappointment, it quickly became clear that the authors, in fact, do not use the entire dataset. Instead, they use minibatches of the dataset (lines 173-179), something that is done in virtually every other deep learning paper (including the ones I mentioned above).

3) Very poor results in CIFAR10/MNIST: I was initially disappointed that when it comes to image datasets, the method is tested only in very simple and long-outdated datasets, like MNIST and CIFAR-10. I would have expected that 'a general framework' that goes 'beyond deep learning' would actually do some more realistic evaluations in more complicated datasets. However, I was even more disappointed to see the results in CIFAR10, where the method reaches only 68.2%. CIFAR10 is a dataset where SOTA is over 99.5% reached by a method that uses Transformers [H]. Even compared to only non-parametric methods (which the method claims to be, although it is a mix between parametric and non-parametric), the results in other works are considerably better (75.89% [I]). Similarly, the method reaches only 98.3% accuracy in MNIST, way below other papers that reach 99.8% (including more than a decade old work of [J]) or the non-parametric method of [I] that reaches 99.44%.

[H] Dosovitskiy et al., An Image is Worth 16x16 Words: Transformers for Image Recognition at Scale, ICLR 2021

[I] Blomqvist et al., Deep Convolutional Gaussian Processes, ECML-PKDD 2019

[J] Cireşan et al., Multi-column Deep Neural Networks for Image Classification, CVPR 2012

Verdict: If the paper would have been presented for what it is: a paper that combines ideas from other fields to progress the task of classification/regression in tabular datasets, this paper would have been an interesting one, and probably deserving of getting published at NeurIPS. However, the claims of the paper are massively exaggerated, the related work does not mention other papers that roughly do the same thing (in more complicated tasks), making the abstract, introduction, and conclusions completely misleading. Adding to that, the weak results in image datasets, I do not see the paper being fit for publishing at the NeurIPS conference. My low score is quite heavily weighted towards the misleading parts of the paper and it promising far more than it shows.

**Update:** Please see the response to rebuttal for the updated version of the review, and why I changed my original score.


**Time Spent Reviewing:**

4 hours

---

> ### Author Response · Authors · 2021-08-09
> **Response to reviewer bbzv**
>
> Dear reviewer bbzv,
>
> We sincerely thank you for your hard work and helpful feedback.
> We ask that you read our comment above, addressed to all reviewers, first.
> Below we address your specific comments as best as we can, and we hope you will engage with us actively during the discussion period to clarify any remaining points.
>
> ## 1. Contributions of NPT
>
> ### NPTs Are Not Metric Learning
>
> We think there has been a misunderstanding: there are major differences between our approach and the related work [A-G] you cite.
> (In the following, we will refer to [A-F] as ‘metric learning’ for convenience.)
> In particular, metric learning does not include learned interactions between training and test data at *test prediction time*, which is a core contribution of our approach – non-parametric predictions *through explicit modelling of interactions between training points and test points*.
> Note that we here refer to training and test data being taken *jointly as input* to the neural network model *at test time* ($f_\theta(\mathcal{D}_\text{train}, \mathcal{D}_\text{test})$), in contrast to *independent embeddings* of datapoints being used at test time in metric learning ($f_\theta(x_\text{test})$, e.g., with $\theta$ the optimum of a triplet loss or other).
>
> > gives the impression that this is the first paper that tries to model some sort of relation between the data
>
> We regret that we conveyed this impression and commit to fixing this in a future draft.
> Note that in our introduction, related work section (in S.3 and Appendix D), and conclusion we cite an extensive list of literature that does model relations between or within data points such as (Deep) Gaussian Processes, Deep Kernel Learning, Neural Processes, Axial Transformers, MSA Transformers, GCN, GAT, GIN, and Neural Relational Inference (see paper for references).
>
> Nevertheless, we stand by our claim in the first sentence that non-parametric prediction – that is, prediction in explicit dependence on training data at test time – is generally uncommon in supervised deep learning settings.
>
> We hope that you engage with us to further discuss similarities and differences of the above approaches with NPTs during the discussion period, and we give some more detailed thoughts on metric learning below.
>
> ### Detailed Comments on the Suggested Related Work
>
> Thank you for suggesting references [A-G].
> We are happy to add a detailed discussion of metric learning/person re-identification to our related work section, and to extend the discussion of Graph Neural Networks (GNNs).
> In particular, we agree that papers [E-G] are important related work through their use of GNNs.
>
> In [A-D], datapoints interact *only during training* and *only through the loss function*, which is defined on multiple samples and encourages the learning of useful representations.
> At test time, the CNN is applied *independently* to the samples, such that these approaches do not leverage *learned interactions* at test time – only a fixed procedure using the datapoint embeddings.
>
> This is in stark contrast to NPTs, which explicitly perform self-attention between training features, training targets, and test features at *test* time.
> This allows NPTs to learn to perform operations such as test-time lookup between training and test inputs.
> For example, NPTs can learn to copy the targets of training points with matching features to the test target.
> This is demonstrated by the semi-synthetic experiments in the paper, especially the intervention experiments in S. 4.2 and extensions in Appendix B.1.2, where we complicate the task further such that it cannot be solved by Gaussian Processes/k-NN (or metric learning).
> None of the suggested related work [A-G] can learn to perform such operations, because they do not consider learned interactions between training and test points at test time and/or do not take training targets as inputs.
>
> Approaches [E-F] apply Graph Neural Networks (GNNs) between samples after embedding datapoints independently.
> However, the most obvious difference to NPTs here is that [E] throws away the GNN after training.
> The GNN only serves to improve the embeddings of the CNN, and at test time, again only the CNN is used, which does not use learned relations between inputs at test time.
> Therefore, at test time [E] is identical in spirit to [A-D] and distinctly different to NPTs.
>
> This is also the main modus operandi for [F].
> However, in their appendix, [F] does explore the targeted construction of test-batches, which to some extent *could* allow them to do things similar to NPTs.
> Even then, unlike NPTs, they explicitly do not include training features in the test-batches, and cannot include training targets (because the architecture does not allow for it).
> Note that taking (known) targets as input to the model at test time is key to some of the intriguing properties of NPT, such as those demonstrated with the semi-synthetic experiments discussed above.
> We further found that our stochastic target masking, which encourages use of input training targets at test time, was indeed beneficial to NPT performance on small data (Appendix C.1.2).
> Apart from being distinct from our work and solving a different problem, we also note that *[F] was published* at ICML 2021, in July, which is significantly *after the NeurIPS submission deadline*.
>
> ### Summary
>
> NPTs use self-attention between training and test datapoints (features and targets) to compute predictions in a non-parametric fashion.
> In contrast, “the objective of deep metric learning (DML) is to learn embeddings that can capture semantic similarity and dissimilarity information among data points” – as is stated in the reference [A].
> We believe that the goals and methods of metric learning and NPTs therefore are distinct.
> We value your opinion, and ultimately hope to resolve any misunderstanding.
> Therefore, we hope to engage with you in the discussion period.
> Particularly, we would be open to a discussion on how we can adjust the paper to avoid further misunderstanding, e.g., by making the first sentence in our abstract clearer or by emphasizing how our learned train-test-interactions at test time are distinctly different from metric learning.
>
> ## 2. Minibatch Approximation
>
> Thank you for drawing our attention to this. There may be a misunderstanding here.
> In fact, we use the minibatch approximation *only* if the full dataset does not fit into memory.
> In 4/10 of our tabular datasets, we do indeed use the full dataset *without requiring minibatches* (see Tables 11 and 12 in the Appendix).
> We regret that l. 62 was perceived as misleading.
> However, we note that on l. 65, directly afterwards, we clarify that “we approximate this where necessary for large data (S. 2.6)”.
> In S. 2.6 (ls. 173-179) and S. 5 we then address the scaling limitations of NPTs for larger datasets explicitly.
> We will revise the draft to mention “minibatches” explicitly in l. 65 (or earlier) to make this immediately clear for future readers.
>
> > they use minibatches of the dataset (lines 173-179), something that is done in virtually every other deep learning paper
>
> While that is true, we stress that the importance of our approach is that we perform attention *between* inputs of the minibatch, including both features and (some) labels at training and test time.
> For us, minibatches are an approximation to the full dataset and not a way of parallelizing model training/prediction.
> Scaling an NPT-like model to take as input a large-scale dataset, followed by a “sophisticated” embedding or memorization as you mention, is important and interesting follow-up work. We point towards this in ls. 352-353 when describing “future work in principled attention approximations, such as learning representative input points, kernelization, or other sparsity-inducing methods”.
>
> ## 3. Results for CIFAR-10
>
> The importance of the CIFAR-10 and MNIST results is to demonstrate that NPT makes use of interactions between datapoints even for the image datasets (S. 4.3/Table 2).
> Our intention was not to claim that the results we get are competitive with state-of-the-art image-classification methods.
> In Appendix B.6, our submission includes a discussion of our setup for image data.
> In particular, we explain why we should not be expected to achieve the same results as the study [H] you cite.
> Among these reasons, we mention that such previous works obtain 90%+ accuracy on images with Transformers by pretraining on “large or huge datasets; for example, ImageNet [22,39,74], ImageNet-21k [65], or JFT-300M, with over 375 million labeled datapoints”.
> They also “use significantly more patches (e.g., 256 in [24]) and use higher resolutions, including during fine-tuning by upscaling”.
> Additionally, please see our comment addressing all reviewers for new and improved results on the image datasets using a CNN encoder.
>
> We hope that our response and the new ablations have adequately addressed what we believe is largely a misunderstanding.
> We would greatly appreciate engaging you during the discussion period to talk through any remaining misunderstandings and make changes necessary for you to no longer view the paper as misleading and instead fitting for publication at NeurIPS.
>
> Thank you,
>
> The Authors

---

> > ### Comment · Reviewer_bbzv · 2021-08-11
> > **Response to rebuttal**
> >
> > I thank the authors for their excellent, factual, and calm response to a relatively harsh review. I think that the authors have clarified many of the points I addressed in my original review. Below, I address some of them in the context of the rebuttal.
> >
> > **NPTs Are Not Metric Learning** - of course, they are not. That was never my intention to claim that NPTs are Metric Learning. Obviously, NPTs are models, metric learning is a subfield. My point was that writing the paper as NPTs are going beyond traditional deep learning (which considers only input-output pairs) methods is a bit misleading, and needs to be toned down. This needs to be written entirely in the context of Transformers, in which case, the claims of the authors would be totally valid.
> >
> > **Detailed Comments on the Suggested Related Work** - my comments here were to just point to a few papers where all the contextual relations in the minibatch are considered, not that they are doing the same thing as the authors. In fact, only papers [E, F] use Transformers, and only F uses the training data in inference (and that in a very rudimentary way, compared to this work which goes quite deeper in that aspect). I also appreciate the authors' honesty in Appendix D about citing many other relevant works that consider the data when it comes to making inference decisions. I echo the thoughts of reviewer m1rj who mentions putting the discussion on graph neural networks in the main paper.
> >
> > **Minibatch Approximation** - despite the authors' response, I still think that the sentence is misleading. Quite clearly, the authors use the entire dataset only when it comes to small datasets. When it comes to large ones, be them tabular datasets like Higgs boson, or image datasets like ImageNet, there is no other way except for minibatch approximation. I fully understand that this is different from minibatch parallelization done in the majority of deep learning works, though it has some similarities to [D, E, F]. Nevertheless, I think that this needs to be clarified better, that the method can use the entire dataset when it is small, otherwise needs to approximate the training set in mini-batches.
> >
> > **Results in image datasets** - I still do not see much value in these experiments. They only show that the method can be used with image datasets, but reach very poor results. Which in other words, it means that it cannot be used with image datasets. At the very least, I think that the authors should complement their results with the ones where they replace the linear encoder with a CNN, where the results become competitive, although far from state-of-the-art.
> >
> > Updated verdict: I like that the authors were able to clarify most of my concerns. In hindsight, I realize that my review was needlessly harsh, and most of the raised issues could have been solved with a tiny rewriting of the paper. I like that the authors are willing to make the recommended changes by me to fix these issues. I wish all the conferences would have this type of reviewing system, that allows longer responses and discussions, which ultimately result in better papers and progress in the field.
> >
> > Other issues addressed by the reviewers are very important to be addressed. I emphasize especially the one for computational complexity. The method is significantly more computationally expensive than various Boosting versions, and this needs to be mentioned in the paper. Nevertheless, considering that this is the first in this line of work, I think that eventually, the researchers building over it will hopefully make the method more efficient (additionally, we all know that deep learning is computationally expensive). I am excited to see the other researchers building over this, and I am curious if we will see this method (or follow-ups) being used in Kaggle competitions where Boosting methods dominate.
> >
> > I said in my original response that sold for what it is, the paper deserves to be published in NeurIPS, and I stand by that. After the rebuttal, I think that my concerns have been addressed, and thus **I am massively increasing my score towards the acceptance regime**. I hope that the authors make the changes in their manuscript to address my and other reviewers' issues (who were extremely informative and detailed). This will ultimately result in a better paper.

---

> > > ### Author Response · Authors · 2021-08-11
> > > **Author response**
> > >
> > > Dear reviewer bbvz,
> > >
> > > > I wish all the conferences would have this type of reviewing system, that allows longer responses and discussions, which ultimately result in better papers and progress in the field.
> > >
> > > We could not agree more. We are glad that you appreciate our extensive response, and we are delighted that we can reach an agreement about our submission.
> > >
> > > Rest assured that we will implement your and other reviewers’ suggestions, such as emphasizing computational complexities, extending the GNN discussion and moving it to the main paper, making sure to state our contributions more precisely, and being more upfront and explicit about the mini-batch approximation.
> > >
> > > Again, thank you for your reviewing service and the participation during the discussion period!
> > >
> > > The Authors

---

### Official Review · Reviewer_miRP · 2021-07-15

**Rating:** 7
**Confidence:** 3

**Summary:**

This paper has the following contributions:
* Propose Non-Parametric Transformers (NPT), a trained transformer applying attention across training data points to make predictions.
* Demonstrate that attention across datapoints is crucial for NPTs.
* Display good results of transformers on tabular datasets.
* Apply a self-supervised objective to tabular data by masking some features.

**Main Review:**

I find this paper very interesting and well written. The proposed model is novel and performs well on tabular data. The authors conduct extensive experiments to test the hypothesis that the network leverage the access to training datapoints and does not only use the model's weights.

It is not easy to place the contribution in the litterature and the authors make a significant effort of exposing related work. I would also relate to [1] where the model attend to a (learned) memory which could be seen in your case as the training dataset itself.

**Precision on the NPT architecture.**
"we alternatingly apply attention between datapoints, and attention between attributes of individual datapoints"
It would be helpful to explicitely state if you are using or not MLPs between these layers as this is standard in transformers.

The attention mechanism between attributes is interesting and could be used to cope with variable number of features or repeated features. Unfortunately, these benefits are annealed by the flattening and use of the $d \cdot e$ size vector for ABD. An ablation of ABA (Attention Between Attributes) would be very informative. I suspect that a simple MLP on the features/channels would work as well.

**Evaluation on Image Data.** The authors are showing "early results" on images that are far from significant. Namely, 68.2% accuracy on CIFAR and 98.3% accuracy on MNIST only shows the reader that NPT can be run on images but this is not a contribution. They should at least include a direct comparison to [2] which achieves 87.1% on CIFAR10 with a small transformer and no pretraining.

I found Appendix C.3 on embedding of tabular features very informative and maybe it could be extended in the main text.

The feature masking objective seems interesting and I am curious if it is crucial for the success of NPT. Could you run an ablation study with $\lambda = 0$?


[1] Augmenting Self-attention with Persistent Memory
Sainbayar Sukhbaatar, Edouard Grave, Guillaume Lample, Herve Jegou, Armand Joulin
https://arxiv.org/pdf/1907.01470.pdf

[2] On the Relationship between Self-Attention and Convolutional Layers
Jean-Baptiste Cordonnier, Andreas Loukas, Martin Jaggi
ICLR 2020


**Time Spent Reviewing:**

6

---

> ### Author Response · Authors · 2021-08-09
> **Response to reviewer miRP**
>
> Dear reviewer miRP,
>
> We sincerely thank you for your hard work and helpful feedback.
> We ask that you read our comment above, addressed to all reviewers, first.
> Below we address your specific comments as best as we can, and we hope you will engage with us actively during the discussion period to clarify any remaining points.
>
> ## Related Work
>
> Thank you for suggesting [1] – we will add a discussion of it in our updated related work section.
>
> ## MLP Between Layers
>
> We do not apply any *extra* MLPs between the ABA and ABD layers.
> However, we do apply an MLP with a residual connection at the end of each ABA/ABD layer, see ‘rFF’ in Eq. 4.
> For ABD, this is applied over the flattened $d \cdot e$ representation with $n$ as the batch dimension, and for ABA it is over $e$ with $n \cdot d$ as the batch dimension(s).
> We hope this clarifies things, and we will update the draft to make this more clear.
>
> ## Ablation of ABA
>
> Thank you for suggesting this.
> In general, we believe that ABA should be a fairly robust and general component of the NPT model that is somewhat well-motivated by the desire to learn complex representations and transformations of the features (ls. 145-147).
> However, you – and m1rj who also suggested this – are absolutely right.
> Practically, an MLP on the flattened representation might suffice – especially on tabular data.
> Following your advice, we have performed this ablation and report results in the main message to all reviewers.
>
> ## Image Data
>
> Thank you for suggesting the comparison to [2]. We will add this to the paper!
> Note that the purpose of our “early results” on images was to show that NPTs continue to rely on interactions between datapoints even for images (S. 4.3/Table 2).
> However, we have now updated and significantly improved the results on the image data following suggestions from [m1rj].
> We give those results in the main comment addressed to all reviewers.
>
> ## Embeddings in Appendix C.3
>
> Thanks for the insight that Appendix C.3 was very informative to you.
> We will happily move C.3 to the main text.
>
> ## No Feature Masking
>
> Disabling feature masking is an interesting idea!
> In general, we would expect feature masking during training to be a meaningful regularizer for “easy” datasets, by making the prediction task more difficult and guarding against memorization.
> A quick clarification: are you suggesting we set $\lambda=0$ (l. 160) but keep $p_\text{feature}=0.15$ (l. 154), i.e. we continue to mask out features but do not backpropagate any losses (resulting in a dropout-like operation on the input)?
> We speculate you might instead prefer us to set $\lambda=0$ *and* $p_\text{feature}=0$, such that there is no feature masking on the input anymore.
> We have performed this ablation and give results in the main comment addressed to all reviewers.
> (We hope that this is your intention, and would be happy to discuss adding additional ablations if we have misinterpreted your intention).
>
> We hope that our response and the new ablations have adequately addressed your concerns.
> We would greatly appreciate it if you could engage with us during the discussion period on any remaining barriers to raising your score and confidence.
>
> Thank you,
>
> The Authors

---

> > ### Comment · Reviewer_miRP · 2021-08-25
> > **Reviewer miRP's answer**
> >
> > Thank you for your answers and for the feature masking loss experiment in the main comment. My concern about experiments on image data is also addressed by the updated results. I do not have any concerns left and thus I increased my score to from 6 to 7.

---

> > > ### Author Response · Authors · 2021-08-25
> > > **Re: Reviewer miRP's answer**
> > >
> > > Dear reviewer miRP,
> > >
> > > Thank you for engaging with us during the discussion period.
> > > We are glad that our rebuttal has addressed your concerns and that you have consequently increased your score.
> > >
> > > The Authors

---

### Official Review · Reviewer_m1rj · 2021-07-16

**Rating:** 8
**Confidence:** 4

**Summary:**

This paper introduces a neural network architecture termed Non-Parametric Transformer (NPT) that uses self-attention (i) between data points and (ii) between features within each data point. The core conceptual contribution is to treat supervised deep learning problems as tasks where the model receives the entire dataset (incl. both labeled and unlabeled data points) as input — in contrast to the common approach of providing data samples independently to a deep learning model. The proposed NPT model is evaluated on tabular datasets where the authors report competitive results compared to boosted decision trees (which represent the state of the art). Applied on MNIST and CIFAR10, the NPT model performs reasonably well. The authors further perform a series of experiments to analyze the learned attention patterns and the importance of self-attention between data points for generalization performance.

**Limitations And Societal Impact:**

The main limitation of the method -- its computational complexity -- is briefly discussed in Section 5. It could be valuable to highlight this limitation more prominently by showing explicit wall clock time and memory usage comparisons against baselines.

The authors discuss societal impacts at an impressive level of detail in the appendix.


**Main Review:**

This is a high-quality paper that is exceptionally well-written, well-structured and very clear in its exposition. The idea is novel and well-positioned against related work (with some exceptions). I think that this paper is relevant for the NeurIPS community and I expect that it will inspire a range of follow-up works.

The idea of using self-attention to take into account all other data points (including their labels, if available) both during training and inference is intriguing, and given the rising popularity Transformer-style architectures in the machine learning community, I expect that this method will spawn lots of interest despite its shortcomings in its current form, such as its computational complexity and its limited usefulness/applicability in any of the studied tasks in the paper.

The experimental evaluation is carried out with great care and the authors come up with several insightful experiments to study whether the attention between datapoint (ABD) mechanism learns to make use of features/labels of similar data points.

Overall, I think that this paper can be accepted and that it would make a great addition to the conference program. I do, however, want to highlight several limitations of the method and issues with the paper below, which I encourage the authors to take into account in the next version of the paper.

The core contribution of the paper is the attention between datapoint (ABD) mechanism and its application for semi-supervised learning in tabular data. While the other model contribution, namely attention between attributes (ABA) is interesting, it is not properly experimentally evaluated (e.g. against an alternative, simpler architecture component, such as an MLP on the flattened representation, or simply the identity, i.e. no module at all). The reader might wonder why this module is introduced in the first place, and its necessity is unclear. I could imagine that this module could simply be left out for the tabular tasks without affecting performance in a significant way. For the visual task, it could likely be replaced with a CNN encoder as the Embedding function (Figure 2b), as done in prior work [1], which would reduce computational complexity and likely not hurt performance. I encourage the authors to revisit the choice of why ABA is introduced in the first place.

One core limitation of the method is its computational complexity. The authors highlight strongly competitive results on tabular data, but this only takes into account the validation error / accuracy. I strongly suspect that the proposed method is by far not competitive with the shown baselines when taking into account computation time and memory requirement. These limitations should be highlighted in the main paper (not in the appendix) by showing a direct comparison in terms of wall clock training/inference time (and memory) to give researchers and practitioners a clear picture about the practicality of the technique. I agree with the authors that alleviating this limitation is a promising avenue for future work.

In terms of related work, I encourage the authors to use the additional page for the camera-ready version (should the paper be accepted) to include the discussion on graph-based semi-supervised learning using graph neural networks in the main paper (instead of the appendix), as the closest relatives to the proposed model come from this research area. This includes a highly-cited paper [1] which similarly introduces an attention mechanism that  aggregates information across data points for semi-supervised learning and few-shot learning in an architecture that is closely related to the one presented here. This paper is currently not cited/discussed, but I think it should be closely contrasted against the proposed technique.

Other suggestions for improvement:
* The CIFAR-10 and MNIST results are currently very difficult to interpret in the absence of any baselines. A comparison against a variant of the model that does not use attention between data points (in the main paper) would significantly strengthen this section.
* The statement that "they can learn to completely ignore other inputs, essentially collapsing into a standard parametric model", in the context of perturbed features (Section 4.3) should be verified, e.g. by comparing the attention weights with and without perturbation of the features.

[1] Garcia & Bruna, Few-Shot Learning with Graph Neural Networks (ICLR 2018)


**Time Spent Reviewing:**

4

---

> ### Author Response · Authors · 2021-08-09
> **Response to reviewer m1rj**
>
> Dear reviewer m1rj,
>
> We sincerely thank you for your hard work and helpful feedback.
> We ask that you read our comment above, addressed to all reviewers, first.
> Below we address your specific comments as best as we can, and we hope you will engage with us actively during the discussion period to clarify any remaining points.
>
> ## MLP instead of ABA
>
> Thank you for suggesting the ablations!
> In general, we believe that ABA should be a fairly robust and general component of the NPT model that is somewhat well motivated by the desire to learn complex representations (ls. 145-147).
> However, you are absolutely right; practically, an MLP on the flattened representation might suffice – especially on tabular data.
> Following your suggestion, we have performed an ablation of NPT without ABA layers, such that only the MLP at the end of ABD can transform features (the ‘rFF’ in Eq. 4).
> We give results in our main comment addressed to all reviewers.
>
> ## CNN Encoder
>
> Replacing our linear patching with a CNN encoder for the image data is a great idea!
> We have also performed this ablation and give results in the main thread to all reviewers.
>
> ## Computational Complexity
>
> We agree that it is useful to extend the discussion on computational complexities and plan to give explicit comparisons in terms of training times, memory requirements, and prediction speed of NPT vs. baselines in the main body of the paper.
>
> As of now, NPT does generally require longer training times than the non-neural baselines.
> For example, for the Protein dataset, our current NPT implementation trains in 11 hours, while all boosting methods finish their runs in less than 1 hour including the hyperparameter tuning.
> The exception to this rule is given by some of the baselines (e.g. random forests) which seem to not scale well to large datasets such as Higgs Boson, which is our largest dataset at 11 million datapoints.
> On Higgs, the NPT run takes 142 h compared to 325 h for random forests.
>
> In terms of memory at prediction time, we suspect that NPT is indeed much more memory-intensive than the baselines, as we try to store as much data as possible in addition to the network weights.
>
> In terms of prediction speed, we suspect that our ability to scale NPT to large batch sizes (e.g., 4096 on the Higgs dataset) might give us an advantage in comparison to those baselines that cannot be parallelized well and/or lack GPU support.
>
> Note that these numbers are to just give a rough indication since we did _not_ optimize the baselines or NPT for memory, training time, or prediction speed. Additionally, comparisons between the CPU-only baselines implemented in C++ and GPU-accelerated PyTorch models will also depend on our specific CPU/GPU choices. Further, by incorporating recent tools for scaling attention, we expect future research to significantly improve on these rough numbers.
>
> We look forward to your feedback and plan to add an extended discussion of the above points.
>
> ## Graph Neural Networks (GNNs) Discussion in Main Text
>
> This is a great suggestion – thank you for bringing this to our attention. We will move the GNN discussion to the main body of the paper, as well as extend the discussion with additional related work including [1].
>
> ## Collapse to Parametric Model
>
> Thank you for bringing this to our attention.
> S.4.3 is a crucial point in the paper, in which we perturb (we call it ‘corrupt’) the features of all other datapoints in the input.
> This allows us to study if the prediction changes at any given input row if we perturb/corrupt the features of all other input rows.
> The results in Table 2 show that for some datasets NPT relies on these other rows, and for other datasets it does not.
> In the latter case, the predictions of NPT depend only on the input row of interest, hence NPT (as a function of its input test point and the train set) ‘collapses’ to parametric prediction which depends only on the input test point.
> We would be interested in discussing additional ways in which we could investigate this, but note that, as far as we know, NeurIPS guidelines discourage the upload of any novel figures during the discussion period.
>
> We hope that our response and the new ablations have adequately addressed your concerns.
> We would greatly appreciate it if you could engage with us during the discussion period on any remaining barriers to raising your score.
>
> Thank you,
>
> The Authors

---

> > ### Comment · Reviewer_m1rj · 2021-08-25
> > **Response to authors**
> >
> > Thank you for your extensive reply to my comments and feedback.
> >
> > The two experiments using CNNs and (per-feature) feed-forward networks (MLPs) are insightful. The experiment with using independent MLPs per feature is different than what I had in mind, but it serves as an insightful ablation. Let me clarify which experiment I believe would be necessary to establish that ABA is indeed a useful component: instead of ABA, one should try flattening the features (per data point) into a single feature vector and pass this through an MLP. This is how tabular / categorical data is typically processed using neural networks and would serve as a more standard way of feature processing. The rFF ablation you're investigating does not allow for feature mixing and hence it is not too surprising that it performs worse than ABA.
> >
> > Overall, I think the paper could be made stronger by showing a stronger experimental justification for ABA or leaving it out altogether. The core contribution in my view lies in the ABD module and in how it is applied for learning with tabular data. I would expect that replacing ABA with an MLP (as suggested above) would a) give better results, and b) lower the computational complexity of the approach.
> >
> > I still think the paper can be accepted with the revisions promised in the rebuttal, since it will likely spur plenty of follow-up research in this direction (especially given the computational complexity of the approach, for which we will likely see follow-up work) and is valuable for the community. I heavily encourage the authors, however, to scrutinize the choice of the ABA module and consider leaving it out altogether.

---

> > > ### Author Response · Authors · 2021-08-25
> > > **Response to reviewer m1rj**
> > >
> > > Dear reviewer m1rj,
> > >
> > > Thank you for your reply and appreciation of our rebuttal.
> > >
> > > We believe there has been a misunderstanding about the specifics of our ‘Ablation 1 – NPT Without Attention Between Attributes (ABA)’.
> > >
> > > The MLP at the end of the each ABD layer _does_ act on the flattened $d \cdot e$ representation of the $n$ datapoints with $n$ as the batch dimension.
> > > (For ABA it is over the embeddings $e$ with $n \cdot d$ as the batch dimension(s).)
> > >
> > > *Therefore, the MLP at the end of ABD has the ability to mix representations between the $d$ features* (cf. ‘rFF’ in Eq. 4 and Eq.5/S. 2.4 in the paper).
> > >
> > > > instead of ABA, one should try flattening the features (per data point) into a single feature vector and pass this through an MLP.
> > >
> > > Leaving out ABA therefore results directly in the setup that you describe here.
> > > As we show, the inclusion of ABA usually improves performance, although there are some datasets for which the MLP-only variant performs better.
> > >
> > > We regret that we have not made this more clear in our original message but hope that our response has adequately addressed your concerns.
> > > Further, we will clarify this in the original draft.
> > > We would greatly appreciate it if you could continue to engage with us during the discussion period on any remaining barriers to raising your score.
> > >
> > > Thank you,
> > >
> > > The Authors

---

> > > > ### Comment · Reviewer_m1rj · 2021-08-25
> > > > **Response to authors**
> > > >
> > > > Thank you for the quick clarification.
> > > >
> > > > I indeed misunderstood your ablation setup and the setup you describe is what I had in mind. This certainly alleviates my concern around the benefits/necessity of ABA. Given that the ABA-free model using CNNs works so well on images, there is still no clear picture as to whether ABA should be presented as an essential component of the architecture (or just as a domain-specific modeling choice), but I do agree that the fully-attentional architecture has a certain appeal.
> > > >
> > > > Given that one of my major concerns was addressed with this ablation (and your clarification), I can confidently raise my score.

---

> > > > > ### Author Response · Authors · 2021-08-25
> > > > > **Re: Response to authors**
> > > > >
> > > > > Dear reviewer m1rj,
> > > > >
> > > > > We are glad we could clear this up and happy to see your score and confidence increase.
> > > > > When adding the ablations to the paper we will make sure to point people towards the option of leaving out ABA layers.
> > > > >
> > > > > Again, thank you for engaging with us during the discussion period.
> > > > >
> > > > > The Authors

---

### Official Review · Reviewer_iTJu · 2021-07-17

**Rating:** 8
**Confidence:** 3

**Summary:**

Deep parametric models have demonstrated tremendous success in NLP, computer vision, and many other settings. These models typically take as input an instance and output a prediction / label(s) for that instance. In this paper, the authors describe an approach that stands in contrast, the input to the model is the _entire_ dataset. Predictions are made collectively via attention between (1) data points and (2) attributes. Unlike non-parametric models that solely interpolate between instances, the proposed Non-Parametric Transformers (NPTs) learn attention mechanisms between data points and between attributes.

The authors describe how the NPT can be trained with a masking approach analogous to masked language-models as well as an approach analogous to 'standard' classification settings.

The authors provide extensive experiments that not only demonstrate the empirical effectiveness on established benchmarks tasks, but also to better understand the behavior and capabilities of each component of the NPT model.

**Limitations And Societal Impact:**

I believe the authors do a good job with this. Perhaps, additional considerations / notes could be added to address how the potential for biases in the model might manifest differently in NPT compared to standard parametric models.

**Main Review:**

The intersection of non-parametric models and deep parametric models is an area that has garnered much attention (and rightfully so, in my opinion). Works such as Neural Processes, Deep Gaussian Processes, and Deep Kernel Learning have had significant impact. The proposed NTP model advances these previous works in significant, not incremental, ways. It provides a novel architecture, which has capabilities not achieved by previous works. In particular, I believe the paper offers the following merits:

* **Innovative Architecture** - The NPT architecture is novel and interesting and offers new perspectives on the ways to design and parameterize deep transformer-like architectures.
* **Inspiration for Future Work** - The paper has the potential to lead to future work both in advancing NPT-like methodologies as well as applications of NPT to a wide array of other tasks.
* **Empirical Analysis** - The authors carefully design experiments to test the performance of the model in different characteristics of the model. These provide insights into how the model behaves in particular circumstances.
* **Well written** - The manuscript is very clearly written and is a pleasure to read. The figures and illustrations are clear and informative. The experimental hypotheses are clearly defined and experiments are illustrative of the merits of the approach. The description of the model is also clear and well motivated.

Overall, I think that this is a very strong paper. Here are some ways in which I think it could be improved:

* **Retrieve and Edit** - There are seemingly some similarities between approaches such as retrieve-and-edit models that retrieve relevant instances and transform them to make a prediction for a target instance, e.g. (Hashimoto et al, 2018), inter-alia. It would be interesting to explore the relationship between such approaches and NPT.
* **Transductive, Case-based, Label Propagation** - Classic approaches such as transductive learning or case-based reasoning I believe are also quite relevant to NPT. Both of these paradigms have also seen a recent revival combining deep learning models and transductive / case-based approaches. It would also be interesting to understand relationships between label propagation models (say on fully connected graphs) and the NPT model.
* **Structured Prediction** - Many structured prediction tasks can naturally extended their associated models / objectives / energy functions to the dataset level, it might be interest to understand relationships between NPT and structured prediction models.
* **Differences from Interpolation** - I am a bit unsure that I follow the argument in lines 170-172: "For example, NPTs could learn to assign test datapoints to clusters of training datapoints, and predict on those points using interpolation of the training targets in their respective cluster. We explore the ability of NPTs to solve such complex reasoning tasks in §4.2." . In 4.2, what I am having trouble with is why this is a reasoning task rather than an interpolation task. Isn't an interpolative to put all interpolation weight on the unmasked 'copy' of the masked instance?
* **Theoretical results** - I don't think this is a requirement of the paper. I think that the empirical analysis is illustrative of the method. The impact of the paper could perhaps be improved with a theoretical analysis of the method.

**Update after author response** Thanks very much to the authors for their response. I believe this is a interesting paper with many notable merits.


**Time Spent Reviewing:**

4

---

> ### Author Response · Authors · 2021-08-09
> **Response to reviewer iTJu**
>
> Dear reviewer iTJu,
>
> We sincerely thank you for your hard work and helpful feedback.
> We ask that you read our comment above, addressed to all reviewers, first.
> Below we address your specific comments as best as we can, and we hope you will engage with us actively during the discussion period to clarify any remaining points.
>
> ## Suggested Related Work
>
> Thank you for suggesting the additional references.
> We will significantly extend our discussion of related work (S. 3, Appendix D), including comparisons to retrieve-and-edit models, transductive and case-based models, label propagation, and structured prediction. We also believe these connections are highly interesting and worth exploring.
> If there are specific references you would like us to mention (in addition to Hashimoto et al., 2018), please list them during the discussion period.
>
> ## Differences to Interpolation
>
> Thank you for bringing this to our attention.
> To make sure we understood your question correctly, could you please clarify if by ‘interpolative’ you are referring to simple non-parametric models, like k-NN and Gaussian processes, that predict by simply ‘interpolating’ from the training data; whereas a reasoning task should require a more complex sequence of operations?
> We feel that the semi-synthetic tasks in S. 4.2 can be called ‘reasoning’ tasks as there exists a distinct sequence of logical/computational ‘reasoning’ operations (compare features, copy and paste target) that leads to a perfect solution that is, in this case, even independent of the input data distribution.
> Further, the extensions to S. 4.2 we present in Appendix B.1.2, where, e.g, a “+1” must be subtracted from the lookup target row, can no longer be solved with a simple ‘interpolative’ model like k-NN/GPs.
> We will clarify our phrasing in ls. 170-172 as well as in S. 4.2 to make this more clear to future readers!
>
> ## Theoretical Results
>
> Thank you for the suggestion of studying the theoretical properties of NPTs further.
> In Appendix A, we study the equivariance properties of NPTs, which we believe to be crucial to their empirical success, and provide a theoretical analysis of the model.
> We would be happy to extend our theoretical analysis further, and would gladly try to incorporate any concrete ideas you might have in mind.
> These would be a good topic to discuss during the discussion period!
>
> ## Biases Manifest Differently in NPT
>
> This is a really interesting point, thanks for bringing it up.
> In our opinion, the fact that the training data is explicitly part of the NPT model (at test time) leads to interesting questions when considering bias.
> In Appendix F, we speculate that NPT might allow for new ways of adjusting for biased/unfair predictions: “Instead of retraining the model to elicit changes in prediction [...] NPT could be ‘prompted’ with a different set of context datapoints to modify its predictive behaviour.”
> Is this similar to what you had in mind?
>
> Thank you,
>
> The Authors

---

> > ### Comment · Reviewer_iTJu · 2021-08-30
> > **Thanks for your response**
> >
> > Thanks very much for your detailed response! And apologies for my delayed one.
> >
> > > Differences to Interpolation
> >
> > Yes, k-nn / GP is what I meant, and table B.1.2 address my confusion precisely. Thank you for this clarification, that result is very convincing.
> >
> > > Biases Manifest Differently in NPT
> >
> > Thanks for this pointer in the appendix as well. Yes, this is what I had in mind.

---

> > > ### Author Response · Authors · 2021-08-31
> > > **Re: Thanks for your response**
> > >
> > > Dear reviewer iTJu,
> > >
> > > thank you for your reply!
> > > We're glad our rebuttal could address your suggestions and will make sure to add the above clarifications to the camera ready version of the paper.
> > >
> > > Very best
> > >
> > > The Authors

---

### Author Response · Authors · 2021-08-09
**Response to all reviewers**

We thank all reviewers for their hard work and detailed feedback.
We are excited about the new conference guidelines encouraging reviewers to have an active discussion with the authors during the discussion period – we believe this will help resolve any misunderstandings.

After reading the below, please see the individual responses where we ask specific questions to clarify ambiguities and address individual reviewers' comments.

We were glad to see that the reviewers feel that our submission is a “very strong paper” (iTJu) that “advances previous works in significant, not incremental, ways” (iTJu), is a “high quality paper” (m1rj) that is a “great addition to the conference program” (m1rj), and is both “novel” (miRP) and “very interesting” (miRP).
Further, we are happy to see you all highlight that our experimental evaluation is “carefully designed” (iTJu), “carried out with great care” (m1rj), “insightful” (m1rj), “extensive” (miRP), and that we “do a good job at giving insights” (bbzv).
We were glad that you feel that our experimental results “demonstrate the empirical effectiveness” (iTJu), show “competitive” (m1rj), “good” (miRP) and “very good” (bbzv) results on the tabular datasets.
Likewise, all reviewers agree that the paper is “very clearly written and is a pleasure to read” (iTJu), “exceptionally well-written, well-structured and very clear in its exposition” (m1rj), “well written” (miRP), and that we “do a good job at explaining” (bbzv).
Lastly, we are particularly excited that [iTJu] and [m1rj] agree with us that this paper “has the potential to lead to future work” (iTJu), “will inspire a range of follow-up works” (m1rj), and “will spawn lots of interest” (m1rj).

## Related Work

We agree with [miRP] that “it is not easy to place the contribution [of NPTs] in the literature”, and we note that our draft currently addresses Deep Non-Parametric Models, Neural Processes, Attention-Based Models, Few-Shot Learning, Meta-Learning, and Prompting in S. 3 as well as Semi-Supervised Learning, Graph Neural Networks, and Tree-Based models in Appendix D.
We are thankful for all related work suggested by the reviewers that covers a wide variety of additional machine learning domains: Retrieve-And-Edit, Transductive Learning, Case-Based Reasoning, Structured Prediction, Metric Learning, and Self-Attention (with memory/on images).
We will carefully incorporate all suggestions to further strengthen our discussion of related work, noting that [m1rj] generally feels that we are “well-positioned against related work” and [miRP] recognizes our “significant effort of exposing related work”.

While we thank [bbzv] for their suggestion of adding metric learning to the related work, we think there has been a misunderstanding when they say that NPT makes no contributions beyond the metric learning and similarity learning literatures: at test time, NPT leverages a learned self-attention mechanism between the test inputs and the *training datapoints (including their labels)*, whereas metric and similarity learning generally rely on independent embeddings of datapoints.

## Additional Experiments

### Summary of Experiments and Results

We conducted three additional experiments suggested by the reviewers: (1) ablating Attention Between Attributes (ABA); (2) ablating feature masking; (3) using a CNN encoder with NPT on CIFAR-10 image classification. **To summarize the results**: (1) inclusion of ABA layers improves performance on 7/10 datasets; (2) feature masking improves performance on 8/9 datasets (one result pending); (3) using a ResNet18 feature encoder, NPT achieves competitive performance with no pretraining (93.7%) and still uses between-datapoint interactions in its predictions, evidenced by a 1.2% decrease in accuracy via a data corruption experiment (S.4.3).
We thank the reviewers for contributing these interesting experiment ideas. We consider all results positive, supporting the inclusion of both ABA layers and feature masking in the base NPT model, along with the applicability of NPT in general settings beyond tabular data.


### Ablation 1 – NPT Without Attention Between Attributes (ABA)

Following helpful feedback from [m1rj] and [miRP] we performed an ablation to test if ABA layers are beneficial in practice.
For this, we leave out the ABA layers as suggested by [m1rj], such that the MLP at the end of the ABD layers (see ‘rFF’ in Eq. 4) is now the only way for the model to independently transform the features of input datapoints.

Our results, given in Table 1 below, show that, generally, ABA is a useful component of the NPT architecture – leaving out ABA increases performance only for 3/10 datasets.
Interestingly, all three of these datasets are regression tasks, which may warrant further investigation.
We observe the largest difference for the Poker Hands dataset, which requires complex reasoning between input features: the ablation only achieves  57.4% accuracy compared to 99.3% for full NPT in the same number of training steps.
These results support our hypothesis that ABA is useful when the dataset requires complex transformations of the features.
Our most general recommendation would be to default to using NPTs with ABA layers, as they boost performance on the majority of datasets we examine.
However, if practitioners can spend the extra compute, exploring NPTs without ABA can be worthwhile.
We will amend the paper with these results and recommendations.

### Ablation 2 – NPT Without Feature Masking

Following feedback from [miRP] we perform an ablation to test if “the feature masking objective [...] is crucial for the success of NPT” (miRP).
For this, we disable all stochastic masking of input features by setting $p_\text{features}=0$ (l. 154).
Our results, also in Table 1 below, show that for 9/10 datasets, enabling feature masking yields at least a small improvement in performance.
On Poker Hands, we observe strong overfitting on the training set (and therefore large generalization error) when disabling feature masking.
Analogous to the first ablation above, our  most general recommendation would be to use NPTs *with* feature masking by default, as it rarely seems to decrease performance and sometimes helps significantly, but to explore NPTs *without* feature masking if computationally feasible.
We will amend the paper with these results and recommendations.

### Improving Results on Image Data

As suggested by [m1rj], we study the following modification to the NPT architecture to increase performance on the image datasets: we replace our linear encoder with a CNN, which is then followed by several rounds of ABD on the CNN encodings.
Here, we choose a ResNet18 as the encoder, followed by 4 blocks of ABD with 8 heads, applied to the CIFAR-10 dataset.

With this architecture we achieve a test accuracy of 93.7%, much improving on the previously reported results. An important question that arises with this experiment is if the between-datapoint interactions still play a role in prediction, even in the presence of the stronger inductive biases of the CNN encoder. To test that, we perform data corruption experiments (S. 4.3) to see to what extent the model uses attention between datapoints in the reported accuracy score:
the data corruption test decreases accuracy by 1.2%, such that we are confident that NPT is again meaningfully relying on other datapoints for prediction.
The ResNet18 alone achieves a test accuracy of 93.9%.

We further note that with a ResNet18 encoder pretrained on ImageNet, our ResNet + NPT architecture achieves a test accuracy of 94.7% on CIFAR10, and loses 0.7% in the row corruption experiment, whereas the pretrained ResNet18 alone achieves a lower 94.2% accuracy. We believe that an exploration of how pretraining might affect the performance of NPT and the extent to which predictions rely on other datapoints is interesting future work.

Altogether, the above presents a big improvement over our previous image classification results and hopefully demonstrates that the ideas of NPTs are useful beyond the tabular domain.

Again, we thank all reviewers for the helpful feedback, suggested related work, and experiments.
We hope to actively engage with you in the discussion period to clarify any remaining points!

Table 1: Additional ablations of the NPT architecture. Experiment protocol identical to the main paper, e.g., results for small datasets $\pm$ standard errors over 10 CV splits.

|                        | 1) NPT without ABA | 2) NPT without feature masking | Default NPT       |
|------------------------|--------------------|--------------------------------|-------------------|
| *Classification*       |                    |                                |                   |
| Poker Hand (Acc ↑)     | $57.4$             | $69.7$                         | $99.3$            |
| Forest Cover (Acc ↑)   | $95.5$             | $96.0$                         | $96.7$            |
| Higgs Boson (AUC ↑)    | $0.859$            | $0.871$                        | $0.892$           |
| Income (AUC ↑)         | $0.952$            | $0.952$                        | $0.952$           |
| Kick (AUC ↑)           | $0.767$            | $0.766$                        | $0.770$           |
| Breast Cancer (AUC ↑)  | $0.992 \pm 0.008$  | $0.996 \pm 0.006$              | $0.997 \pm 0.001$ |
| *Regression*           |                    |                                |                   |
| Boston Housing (RMSE ↓)| $3.22 \pm 0.25$    | $3.18 \pm 0.35$                | $2.92 \pm 0.15$   |
| Yacht (RMSE ↓)         | $1.15 \pm 0.11$    | $0.50 \pm 0.06$                | $1.27 \pm 0.15$   |
| Concrete (RMSE ↓)      | $4.79 \pm 0.12$    | $5.37 \pm 0.20 $                           | $5.21 \pm 0.20$   |
| Protein (RMSE ↓)       | $3.29$             | $3.59$                         | $3.41$            |

[edit]: Last missing experiment completed and added to table 1.

---

### Decision · Program_Chairs · 2021-09-27

**Decision:**

Accept (Poster)

**Comment:**

This paper explores alternative modeling paradigms for supervised deep learning. In this paper, the authors describe an approach wherein input to the model is the entire dataset along with the query instance. Predictions are made collectively via attention between (1) data points and (2) attributes. This model which attends over entire dataset for predication is named Non-Parametric Transformers (NPTs). The authors describe how the NPT can be applied in various settings, such as classification. Through extensive experiments and ablation studies, the effectiveness of NPT is demonstrated on established benchmarks tasks. Moreover, I (and all the reviewers) found the paper well-written and very clear in its exposition and idea presentation.

We thank the reviewers and authors for engaging in an active discussion, which resulted in clearing a lot of the concerns (e.g. speed/flops) and a lot of constructive feedback were provided to improve the paper. The authors provided extensive new empirical results/ablation studies as part of the discussion, please include them in the final version of the paper as they add great value and understanding to the model as a whole. Overall reviewers reached to the consensus that the paper has many merits and should be accepted.